# The inner nuclear membrane protein Lem2 coordinates RNA degradation at the nuclear periphery

Lucía Martín Caballero[1,2], Matías Capella [1,8], Ramón Ramos Barrales [1,9], Nikolay Dobrev[3,10], Thomas van Emden [1,2], Yasuhiro Hirano [4], Vishnu N. Suma Sreechakram[1,5], Sabine Fischer-Burkart[1], Yasuha Kinugasa[4,11], Alicia Nevers[6,12], Mathieu Rougemaille [6], Irmgard Sinning [3], Tamás Fischer [3,7], Yasushi Hiraoka [4] and Sigurd Braun [1,2,5 ✉]

Transcriptionally silent chromatin often localizes to the nuclear periphery. However, whether the nuclear envelope (NE) is a site for post-transcriptional gene repression is not well understood. Here we demonstrate that *Schizosaccharomyces pombe* Lem2, an NE protein, regulates nuclear-exosome-mediated RNA degradation. Lem2 deletion causes accumulation of RNA precursors and meiotic transcripts and de-localization of an engineered exosome substrate from the nuclear periphery. Lem2 does not directly bind RNA but instead interacts with the exosome-targeting MTREC complex and its human homolog PAXT to promote RNA recruitment. This pathway acts largely independently of nuclear bodies where exosome factors assemble. Nutrient availability modulates Lem2 regulation of meiotic transcripts, implying that this pathway is environmentally responsive. Our work reveals that multiple spatially distinct degradation pathways exist. Among these, Lem2 coordinates RNA surveillance of meiotic transcripts and non-coding RNAs by recruiting exosome co-factors to the nuclear periphery.

Eukaryotic genomes are pervasively transcribed, giving rise to sense and antisense RNAs from intra- and intergenic regions and repetitive elements. Accumulation of cryptic unstable transcripts (CUTs) may cause genome instability through RNA-DNA hybridization[1]. Other coding and non-coding RNAs are continuously transcribed but function during specific developmental stages; for example, fission yeast meiotic mRNAs are expressed but rapidly degraded in vegetative cells[2–5]. Constant surveillance is required to prevent the aberrant accumulation of these transcripts, which is mediated by both nuclear and cytosolic pathways. However, how RNA-degradation pathways are coordinated in the nucleus is not well understood. Although the role of perinuclear anchoring in transcriptional silencing has been intensely studied, the effect of nuclear organization on post-transcriptional repression remains unclear.

Eukaryotic nuclear RNA degradation is mediated by the nuclear exosome, a multiprotein complex containing two $3' \rightarrow 5'$ exoribonucleases, Rrp6 (ribosomal RNA processing 6) and Dis3/Rrp44 (ref. [6]). RNA-surveillance pathways contribute to substrate specificity and RNA processing, assisting exosome-targeting complexes through polyadenylation and RNA helicase activities that unwind RNA secondary structures[6]. In *Schizosaccharomyces pombe*, these targeting complexes include TRAMP (Trf4/5–Air1/2–Mtr4 polyadenylation) and MTREC (Mtl1–Red1 core). The TRAMP complex comprises a non-canonical poly(A) polymerase (Cid14; Trf4/Trf5 in *Saccharomyces cerevisiae*), a zinc-knuckle RNA-binding protein (Air1), and an RNA helicase (Mtr4). Fission yeast TRAMP degrades transcripts derived from pericentromeric repeats and plays a minor role in CUT elimination and small nucleolar RNA (snoRNA) processing[7–9]. The MTREC complex, also known as NURS (nuclear RNA silencing), comprises the Mtr4-like helicase Mtl1 and the zinc-finger domain protein Red1 (RNA elimination defective 1). MTREC mediates turnover of CUTs and meiotic and non-spliced transcripts[9–11] and is orthologous to the human PAXT (poly(A) tail exosome targeting) complex[12]. MTREC assembles into an 11-subunit 'super complex', in which Red1 scaffolds different submodules and recruits Rrp6 via Mtl1 (refs. [10,11,13]). The submodules have different activities and include the canonical Poly(A) polymerase Pla1 and the complexes Red5–Pab2–Rmn1, Ars2–Cbc1–Cbc2, and Iss10–Mmi1 (refs. [9,14–19]).

Exosome targeting and function are best understood in the context of meiotic transcript turnover in *S. pombe*. The YTH (YT521-B homology) protein Mmi1 recognizes hexanucleotide motifs (UNAAAC) known as DSR (determinant of selective removal) sequences[2,3,20]. Substrate binding requires Mmi1 dimerization and interaction with its partner Erh1 (enhancer of rudimentary

[1]BioMedical Center (BMC), Division of Physiological Chemistry, Faculty of Medicine, LMU Munich, Planegg-Martinsried, Germany. [2]International Max Planck Research School for Molecular and Cellular Life Sciences, Planegg-Martinsried, Germany. [3]Heidelberg University Biochemistry Center (BZH), Heidelberg, Germany. [4]Graduate School of Frontier Biosciences, Osaka University, Suita, Japan. [5]Institute for Genetics, Justus-Liebig-University Giessen, Giessen, Germany. [6]Université Paris-Saclay, CEA, CNRS, Institute for Integrative Biology of the Cell (I2BC), Gif-sur-Yvette, France. [7]The John Curtin School of Medical Research, The Australian National University, Canberra, Australian Capital Territory, Australia. [8]Present address: Instituto de Agrobiotecnología del Litoral, CONICET, Universidad Nacional del Litoral, Santa Fe, Argentina. [9]Present address: Centro Andaluz de Biología del Desarrollo (CABD), Universidad Pablo de Olavide-Consejo Superior de Investigaciones Científicas-Junta de Andalucía, Seville, Spain. [10]Present address: European Molecular Biology Laboratory, Hamburg Unit, Hamburg, Germany. [11]Present address: Regulation for intractable Infectious Diseases, National Institutes of Biomedical Innovation, Health and Nutrition, Ibaraki, Japan. [12]Present address: University Paris-Saclay, INRAE, AgroParisTech, Micalis Institute, Jouy-en-Josas, France. ✉e-mail: Sigurd.Braun@gen.bio.uni-giessen.de

homolog 1) to form the tetrameric Erh1–Mmi1 complex (EMC). EMC sequesters RNA substrates, preventing their nuclear export and translation[21–25]. Mmi1 further binds the Ser- and Pro-rich protein Iss10 (also called Pir1), which bridges the Mmi1-MTREC interaction[9,11,13,26]. Several DSR-containing genes are marked by histone H3 lysine 9 methylation (H3K9me) in an Mmi1-, Red1-, and Rrp6-dependent manner, potentially adding another layer of control[5,10]. During Rrp6 inactivation or stress exposure, an alternative degradation pathway using RNA interference (RNAi) results in H3K9me deposition at several loci known as HOODs (heterochromatin domains)[27].

In vegetative cells, exosome factors assemble into one or several nuclear foci. EMC-containing foci require the association of Mmi1 with Erh1 (refs. [24,28]), whereas Iss10 is critical for Red1 foci formation and MTREC co-localization with EMC[13,26,28]. During meiosis, Iss10 becomes unstable and Red1 foci disappear[13,29]. This causes Mmi1 to dissociate from MTREC and collapse into the Mei2 dot, a nuclear structure comprising the RNA-binding protein Mei2 and the non-coding RNA *sme2* (also called *meiRNA*)[2,30]. Together, Mei2 and *meiRNA* sequester Mmi1 at the genomic *sme2+* locus, resulting in the inactivation of Mmi1-dependent RNA elimination[30–32]. Iss10 degradation and the disassembly of nuclear foci coincide with the accumulation of meiotic transcripts[29]. It has therefore been proposed that Iss10-dependent foci represent specific RNA-degradation sites[26,28].

Transcriptionally silent chromatin and repressive histone-modifying enzymes are often sequestered at the nuclear periphery[33]. The methyltransferase Clr4 marks perinuclear heterochromatin with H3K9me[34], which is recognized by HP1 (heterochromatin protein 1) chromodomain proteins[35,36]. This heterochromatic platform recruits the Snf2-like/HDAC-containing repressor complex SHREC, restricting access to RNA polymerase II (Pol II)[37]. Several NE proteins contribute to silencing and perinuclear localization of heterochromatin[38–40]. Lem2 is a conserved integral protein of the inner nuclear membrane (INM) containing the LEM (LAP2, emerin, MAN1) and MSC (MAN1-Src1p C-terminal) nucleoplasmic domains. Although the LEM domain contributes to centromere tethering, the MSC domain mediates heterochromatin silencing[38,41–43]. Lem2 promotes recruitment of SHREC to heterochromatin, thus linking heterochromatin silencing and nuclear organization[38]. Whether Lem2 contributes to other modes of gene regulation remains unknown.

Here, we uncover a global role for Lem2 in repressing non-coding RNAs and meiotic genes, which is distinct from its function in heterochromatin silencing. We show that Lem2 cooperates with the nuclear exosome and physically interacts with the MTREC subunit Red1 and its human homolog. Importantly, Lem2 is critical for the perinuclear localization of exosome substrates and their recognition by Mmi1 and Red1. Lem2-assisted RNA targeting is largely independent of Iss10 and Erh1 and their

assembly into nuclear foci. Furthermore, Lem2 contributes to CUT degradation and snoRNA 3′ processing. Altogether, our data imply that multiple RNA-degradation pathways exist and localize to distinct subnuclear structures. We propose that Lem2 supports RNA surveillance by coordinating the degradation of exosome targets at the nuclear periphery.

## Results

**Lem2 mediates repression of non-coding RNAs and meiotic genes.** The INM protein Lem2 anchors and silences constitutive heterochromatin[38,43]. To examine whether Lem2 regulates gene expression through additional mechanisms, we performed transcriptome analysis by RNA sequencing (RNA-seq). A substantial portion of transcripts were upregulated in cells deleted for *lem2+* (*lem2Δ*), whereas only a few were decreased (838 versus 35, from a total of 6,642 transcripts; Fig. 1a,b). The *S. pombe* genome contains roughly 70% protein-coding and 21% non-coding genes[44]. Non-coding RNAs (ncRNAs) were significantly over-represented (61%) among the upregulated transcripts in *lem2Δ* cells (Fig. 1a,c). Upregulated ncRNAs include *sme2*, a key player during meiosis[45], and the snoRNA *sno20* (Fig. 1d). Consistently, the Analysis of Gene Lists (AnGeLi) tool[46] revealed 'ncRNA' as the group of genes most significantly altered in *lem2Δ* cells and also uncovered other genes linked to meiosis and sporulation (Fig. 1e and Supplementary Data 1). Several long terminal repeat (LTRs) transcripts were also increased (Fig. 1a,c). We further examined mutants of Man1 and Ima1, two other integral envelope proteins known to interact with chromatin[47]. Genome-wide analysis of *man1Δ* cells revealed no major transcriptional changes (Extended Data Fig. 1a). Reverse transcription followed by quantitative PCR (RT–qPCR) confirmed that selected meiotic genes and ncRNAs were upregulated in *lem2Δ* cells; in contrast, these were largely unaltered in *man1Δ* and *ima1Δ* cells (Fig. 1f and Extended Data Fig. 1d), indicating a specific role for Lem2 in the regulation of these transcripts.

Since Lem2 represses transcription by recruiting SHREC to constitutive heterochromatin[38], we examined whether heterochromatic factors also regulate meiotic transcripts and ncRNAs. RNA-seq in cells lacking the H3K9 methyltransferase Clr4 revealed increased levels of pericentromeric ncRNAs and subtelomeric mRNAs, as previously shown[48,49] (Extended Data Fig. 1b). Although many of these transcripts were also increased in *lem2Δ* cells, the number of upregulated transcripts was significantly lower in *clr4Δ* cells than in *lem2Δ* cells (103 versus 838, Extended Data Fig. 1b,c). We examined selected targets by RT–qPCR in *clr4Δ* and other mutants deficient in heterochromatin assembly (*swi6Δ*), RNAi (*ago1Δ*), and SHREC (*clr1Δ, clr2Δ, clr3Δ*). In stark contrast to heterochromatic transcripts, *sme2* and *sno20* transcript levels were unaltered in these mutants (Fig. 1g and Extended Data Fig. 1e); *ssm4* showed modest upregulation in *clr4Δ* and *ago1Δ* mutants (Extended Data Fig. 1e),

**Fig. 1 | Lem2 represses ncRNAs and meiotic genes. a**, Volcano plot depicting RNA-seq data from *lem2Δ* versus WT cells. Genes significantly up- (red) or downregulated (blue) are highlighted (log$_2$(fold change) >1 or < –1 with *P* adjusted value < 0.01 by the Wald test, as implemented within the DESeq2 framework). Prominent transcripts are in bold. **b**, MA-Plot of *lem2Δ* cells relative to WT. *x* and *y* axes show the log$_2$(mean expression) and log$_2$(fold change), respectively, in *lem2Δ* over WT. **c**, Pie charts showing distributions of ncRNA, LTR, protein-coding, and other transcripts (pseudogene, rRNA, snoRNA, snRNA, tRNA). Left, genome-wide distribution of transcript features in a WT genome. Right, transcript feature distribution of the significantly upregulated transcripts in *lem2Δ* cells (log$_2$(fold change) > 1 with *P* adjusted value < 0.01 by the Wald test). **d**, Coverage plots showing upregulated transcripts in *lem2Δ* cells from three independent biological replicates. Reads are presented as counts per million (CPM). Genomic coordinates are shown in base pairs (bp). **e**, Table with selected results from gene list enrichment analysis of *lem2Δ* mutants. The AnGeLi tool with a two-tailed Fisher's exact test and a false-discovery rate of 0.05 was used for this analysis[46]. GO B.P., Gene Ontology biological process; reg. of transc. during mei., regulation of transcription during meiosis; mei., meiotic. **f**, Top, domain structures of Lem2, Man1, and Ima1 (length, amino acids). Bottom, transcript levels of *sme2* and *sno20*, quantified by RT–qPCR (*n* = 4 independent biological replicates). **g**, *sme2*, *sno20*, and *tlh1* transcript levels, quantified by RT–qPCR (*n* = 4 independent biological replicates; except *clr2Δ, ago1Δ*: *n* = 3; HC, heterochromatin). **h**, ChIP–qPCR analysis of Pol II-S5P enrichment at *sme2+*, *sno20+*, and *tlh1+* genes (*n* = 3 independent biological replicates). For RT experiments in **f** and **g**, data are normalized to transcript levels of *act1* or the average of selected euchromatic genes (*act1+, tef3+, ade2+*), respectively, and shown relative to WT. For **h**, ChIP data are divided by the input and normalized to the average level of selected euchromatin loci (*act1+, tef3+, ade2+*). For **f–h**, the individual replicates are shown in floating bar plots and the line depicts the median.

in agreement with its location within a heterochromatin island[5]. These data suggest that Lem2 regulates the expression of these exosome targets largely independently of heterochromatin formation.

Although heterochromatin is controlled at the transcriptional and post-transcriptional levels, meiotic genes are mainly regulated through RNA degradation by the nuclear exosome[2,3,20,50]. To determine

whether Lem2 acts transcriptionally or post-transcriptionally, we performed chromatin immunoprecipitation followed by qPCR (ChIP–qPCR) with Ser5-phosphorylated RNA polymerase II (Pol II-S5P). As expected, $clr4\Delta$ cells showed strong enrichment of Pol II-S5P and increased transcription of $tlh1^+$, a subtelomeric gene within heterochromatin (Fig. 1g,h). We also observed moderate Pol

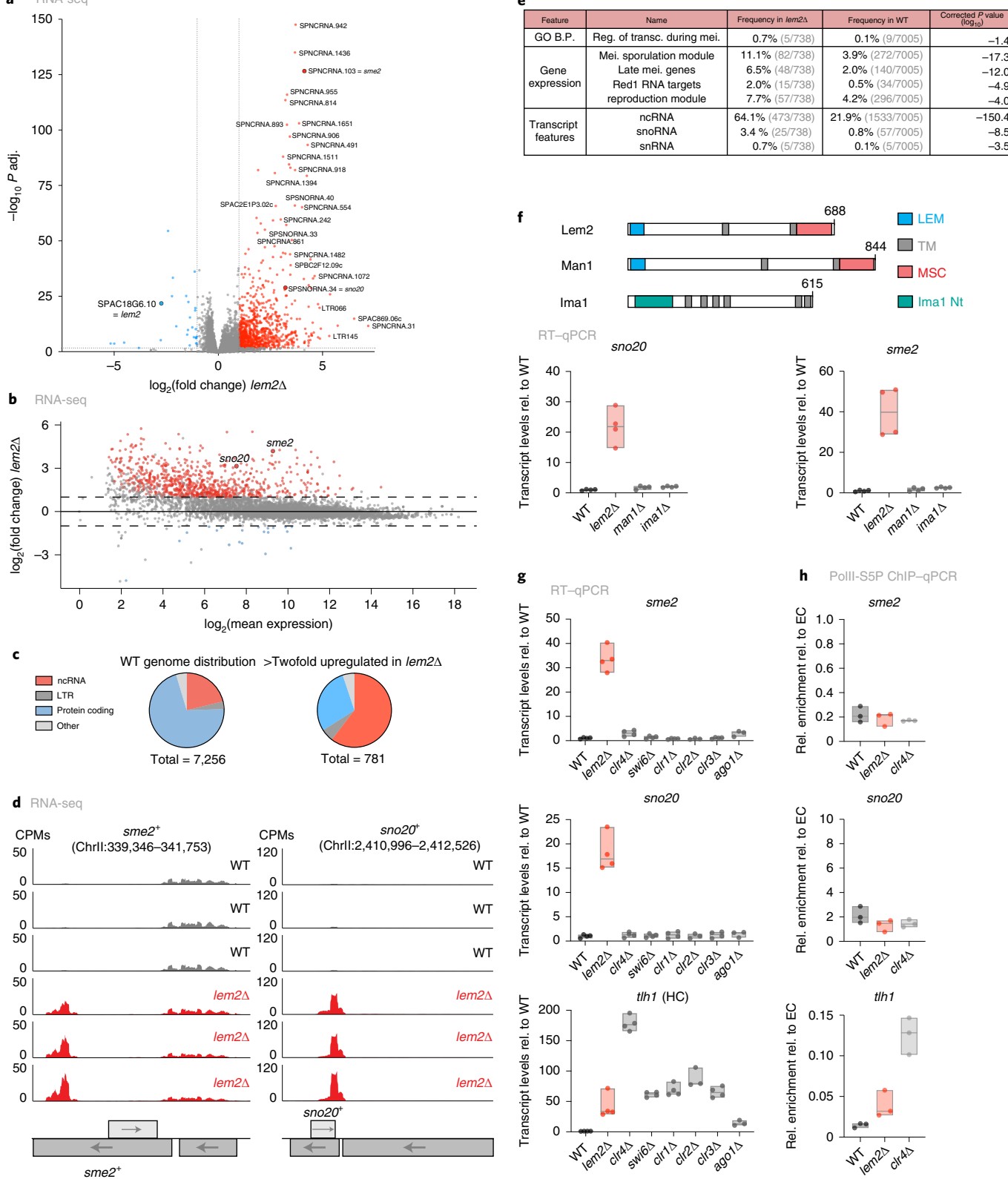

**e**

| Feature | Name | Frequency in $lem2\Delta$ | Frequency in WT | Corrected $P$ value ($\log_{10}$) |
|---|---|---|---|---|
| GO B.P. | Reg. of transc. during mei. | 0.7% (5/738) | 0.1% (9/7005) | −1.4 |
| Gene expression | Mei. sporulation module | 11.1% (82/738) | 3.9% (272/7005) | −17.3 |
| | Late mei. genes | 6.5% (48/738) | 2.0% (140/7005) | −12.0 |
| | Red1 RNA targets | 2.0% (15/738) | 0.5% (34/7005) | −4.9 |
| | reproduction module | 7.7% (57/738) | 4.2% (296/7005) | −4.0 |
| Transcript features | ncRNA | 64.1% (473/738) | 21.9% (1533/7005) | −150.4 |
| | snoRNA | 3.4 % (25/738) | 0.8% (57/7005) | −8.5 |
| | snRNA | 0.7% (5/738) | 0.1% (5/7005) | −3.5 |

II-S5P enrichment at *tlh1*+ in *lem2*Δ cells, consistent with its heterochromatin function[38]. However, Pol II-S5P abundance was unaltered at the *sme2*+ and *sno20*+ loci in *lem2*Δ cells (Fig. 1g,h). This implies that Lem2 regulates meiotic and non-coding transcripts through a transcription-independent mechanism distinct from its role in heterochromatin silencing.

**Lem2 and the nuclear exosome cooperate in RNA surveillance.** Since meiotic transcripts and ncRNAs are major nuclear exosome substrates[9], we examined whether Lem2 mediates post-transcriptional regulation through RNA degradation. We performed RNA-seq in mutants lacking components of the nuclear exosome pathway, i.e., Rrp6 (nuclear exosome), Red1 (MTREC), Erh1 (EMC complex), Iss10 (MTREC/EMC-bridging factor), Ccr4 (CCR4–NOT complex), or Air1 (TRAMP) (Fig. 2a). Principal component analysis (PCA) revealed mutant-specific groups displaying high reproducibility across independent biological replicates (Extended Data Fig. 2a). Differential expression analysis followed by unsupervised *K*-means clustering revealed a striking similarity between transcriptome profiles of *lem2*Δ, *rrp6*Δ, and *red1*Δ mutants. However, *lem2*Δ showed only limited transcriptome overlap with *erh1*Δ, *iss10*Δ, *air1*Δ, and *ccr4*Δ (Fig. 2b). Pairwise transcriptome comparisons revealed strong positive correlations for *lem2*Δ with *rrp6*Δ and *red1*Δ (*R* = 0.79 and 0.65, respectively; Fig. 2c and Extended Data Fig. 2b) and a weaker correlation with *air*Δ (*R* = 0.53; Extended Data Fig. 2c). No correlation was seen with *erh1*Δ or *ccr4*Δ at the genome-wide level (Extended Data Fig. 2c). We also found a strong accumulation of antisense transcripts in *lem2*Δ cells, similar to findings in *rrp6*Δ cells (Extended Data Fig. 2d). Using AnGeLi[46], we analyzed the top clusters containing upregulated transcripts in *lem2*Δ, *rrp6*Δ, and *red1*Δ strains (clusters 1–5). Cluster 1 was enriched for features related to early meiosis (63% frequency) and Red1-mediated degradation (30%). Many of these transcripts were also increased in cells lacking Erh1, consistent with its function in binding to exosome substrates as part of the EMC (Fig. 2b and Extended Data Fig. 2e). In contrast, transcripts present in clusters 2 to 5 were predominantly enriched for ncRNAs (between 53% and 69%), or genes related to splicing, stress regulation, and late meiosis. For cluster 3, we observed an overlap with transcripts specifically increased in *air1*Δ (Fig. 2b and Supplementary Data 2). Together, these results suggest that Lem2 cooperates with distinct exosome factors to control transcripts through multiple degradation pathways.

To confirm co-regulation of these transcripts by Lem2 and the exosome, we analyzed a subset of transcripts recognized by Mmi1 (the 'Mmi1 regulon')[51]. In accordance with transcriptomics (Fig. 2b), RT–qPCR revealed significant changes in Mmi1-regulated RNAs in *lem2*Δ, *rrp6*Δ, and *red1*Δ cells and cells lacking the poly(A)-binding protein Pab2 (Fig. 2d,e). Conversely, transcript levels of *rrp6* and

other exosome genes were largely unaltered in the *lem2*Δ mutant. Therefore, upregulation of exosome targets is not due to loss of exosome gene expression (Extended Data Fig. 2f).

Red1 associates with chromatin at Red1-dependent heterochromatin (HC) islands[5]. We noticed that genes upregulated in *lem2*Δ, such as *mei4*+, are often part of these HC islands, whereas Red1-independent HC islands were mostly unaffected by *lem2*Δ (with the exception of a ncRNA and an LTR gene; Extended Data Fig. 3a). This prompted us to examine the chromatin environment of *mei4*+. Deleting *iss10*+ decreased Red1 binding, as expected. However, we observed no change in chromatin association of Red1 or Mmi1 in *lem2*Δ cells (Extended Data Fig. 3b,c). Moreover, this locus retained H3 dimethylation at K9 (H3K9me2) in *lem2*Δ cells, but not in *rrp6*Δ cells (Extended Data Fig. 3d). We also examined H3K9me at HOODs, which assemble heterochromatin upon loss of Rrp6 (ref. [27]), but observed no H3K9me2 increase in *lem2*Δ cells (Extended Data Fig. 3d). These findings further support the hypothesis that Lem2 acts at the post-transcriptional level to control exosome targets, rather than through chromatin changes.

The close resemblance of transcriptional profiles indicates that Lem2 may act through a pathway that also involves Pab2, Red1, and Rrp6. We examined prominent exosome targets (*sme2*, *ssm4*, *sno20*, and *snR42*) in single and double mutants lacking these factors in combination with *lem2*Δ by RT–qPCR. Although *pab2*Δ, *red1*Δ, and *rrp6*Δ single mutants displayed higher transcript levels of *sme2* and *ssm4* than did the *lem2*Δ mutant (Fig. 2f and Extended Data Fig. 3e), additional deletion of *lem2*+ resulted in a non-additive phenotype, implying an epistatic interaction. In contrast, snoRNA levels showed an additive increase in *lem2*Δ *red1*Δ double mutants (Extended Data Fig. 3e). SnoRNAs are highly abundant and derived from Pab2-mediated 3′ processing of RNA precursors, which accumulate in the absence of Rrp6 (ref. [15]). Consistent with a role for Lem2 in 3′ processing, we found that the precursors of *sno20* and *snR42*, but not their mature forms, accumulated in *lem2*Δ cells (Fig. 2g,h). Together, these results indicate that Lem2 plays a broad role in RNA degradation by collaborating with distinct pathways with different substrate specificities.

**Lem2 interacts with exosome-targeting factor Red1.** Given the epistatic interaction between *lem2*+ and *red1*+ (Fig. 2f), we tested for physical interaction between Lem2 and Red1 by expressing Lem2 fused to green fluorescent protein (Lem2-GFP) and Red1 tagged with 6 copies of the hemagglutinin epitope (Red1-6×HA) from their endogenous loci, as described previously[28,38]. Co-immunoprecipitation (coIP) revealed that Red1 associates with Lem2 in vivo (Fig. 3a). This association was insensitive to RNase and benzonase treatment, similar to MTREC interactions with other factors[9,28].

**Fig. 2 | Lem2 collaborates with the nuclear exosome. a**, Scheme highlighting the main players in the nuclear exosome pathway, with different targeting/bridging complexes (MTREC/NURS, EMC, CCR4–NOT, and TRAMP) providing substrate specificity. Mmi1 recognizes DSR motifs in meiotic genes. Heterochromatic domains (HOODs) are partially controlled through CCR4–NOT and TRAMP. **b**, *K*-means clustering of RNA-seq data, based on differential expression. Clusters with most upregulated transcripts in *lem2*Δ (clusters 1–5) were analyzed with AnGeLi with a two-tailed Fisher's exact test and a false-discovery rate of 0.05 (ref. [46]). The red–blue color scale represents log₂(fold change expression) relative to the WT level for each given gene and mutant. **c**, Scatterplot of genome-wide log₂(fold change) expression from transcripts in *rrp6*Δ versus *lem2*Δ, both relative to WT. The linear regression and the Pearson correlation coefficient (*R*) are shown. **d**, Expression changes in selected genes regulated by the nuclear exosome analyzed by RT–qPCR ('Mmi1 regulon' and others). HC, heterochromatin controls; EC, euchromatin controls. Color scale represents log₂(fold change expression) relative to WT for each given gene and mutant. **e**, Clustering based on Pearson's correlation coefficient (PCC) of RT–qPCR data, with genes regulated by the exosome in the indicated strains. **f**, *sme2* and *ssm4* transcript levels quantified by RT–qPCR. Data are normalized to *act1* transcript levels and are shown relative to WT on a log₂ scale (*n* = 6 independent biological replicates). Letters denote groups with significant differences from one-way analysis of variance (ANOVA) followed by Tukey's post hoc tests at *P* < 0.05. **g**, *sno20* and *snR42* transcript levels quantified by RT–qPCR with primers specific to either the precursor or mature forms. Data are normalized to *act1* transcript levels and shown relative to WT (*n* = 4 independent biological replicates). **h**, Northern blot analyses of RNAs in the indicated strains. Asterisks denote the precursor species of RNAs. rRNA served as a loading control. Two independent experiments were performed with similar results. For **f** and **g**, individual replicates are shown in a floating bar plot, and the line depicts the median.

Lem2 has distinct structural domains that mediate different functions. The LEM domain contributes to centromere association, whereas the MSC domain mediates heterochromatin silencing[38]. In addition, a region adjacent to the first transmembrane domain interacts with the integral membrane protein Bqt4 (refs. [42,52]).

Although deleting the amino terminus did not impair Red1 association, removing the carboxy-terminal MSC domain abolished Red1 binding (Extended Data Fig. 4a). To test whether the MSC domain is sufficient and essential for Red1 association, we performed yeast two-hybrid (Y2H) assays. Consistent with our coIP

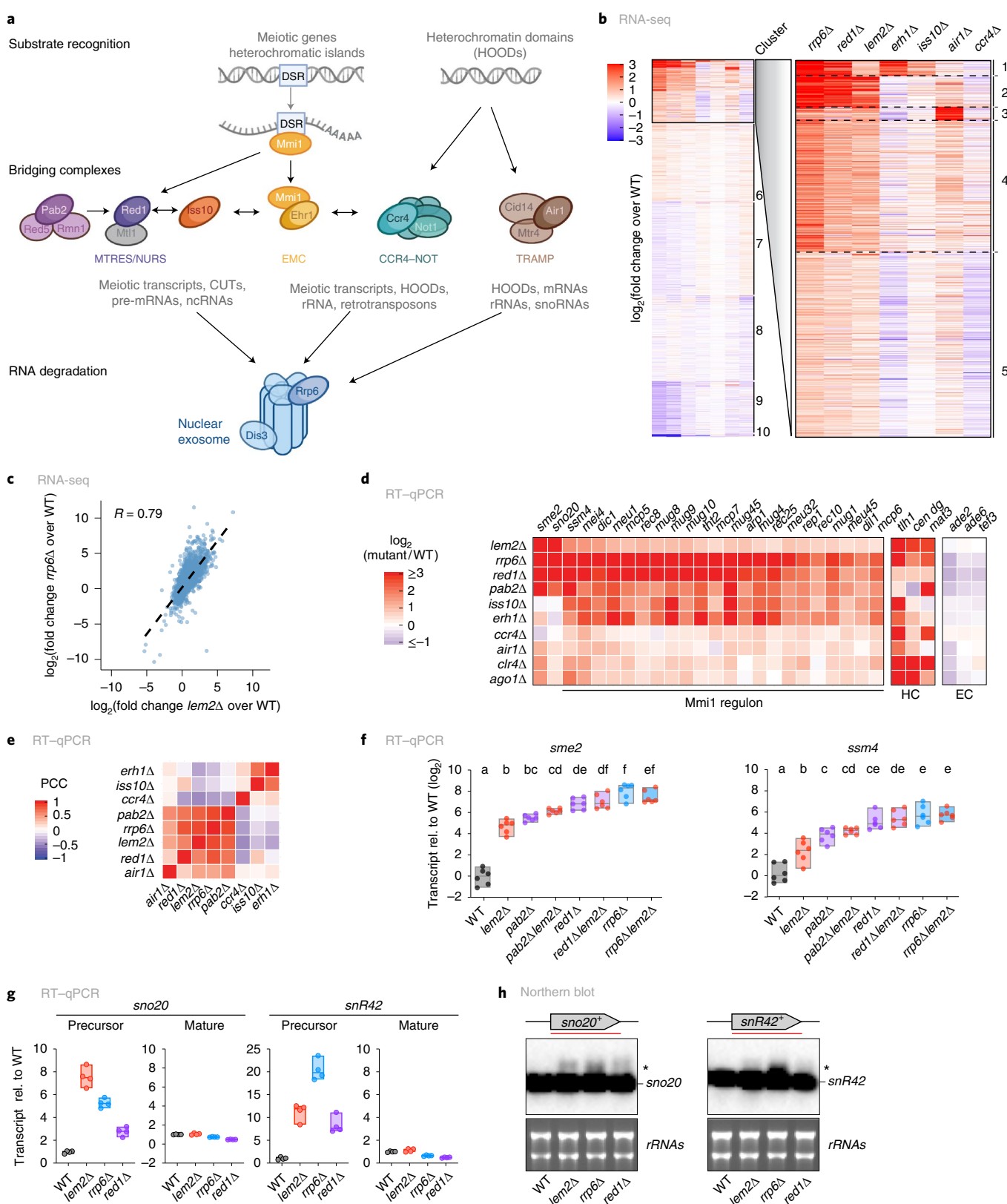

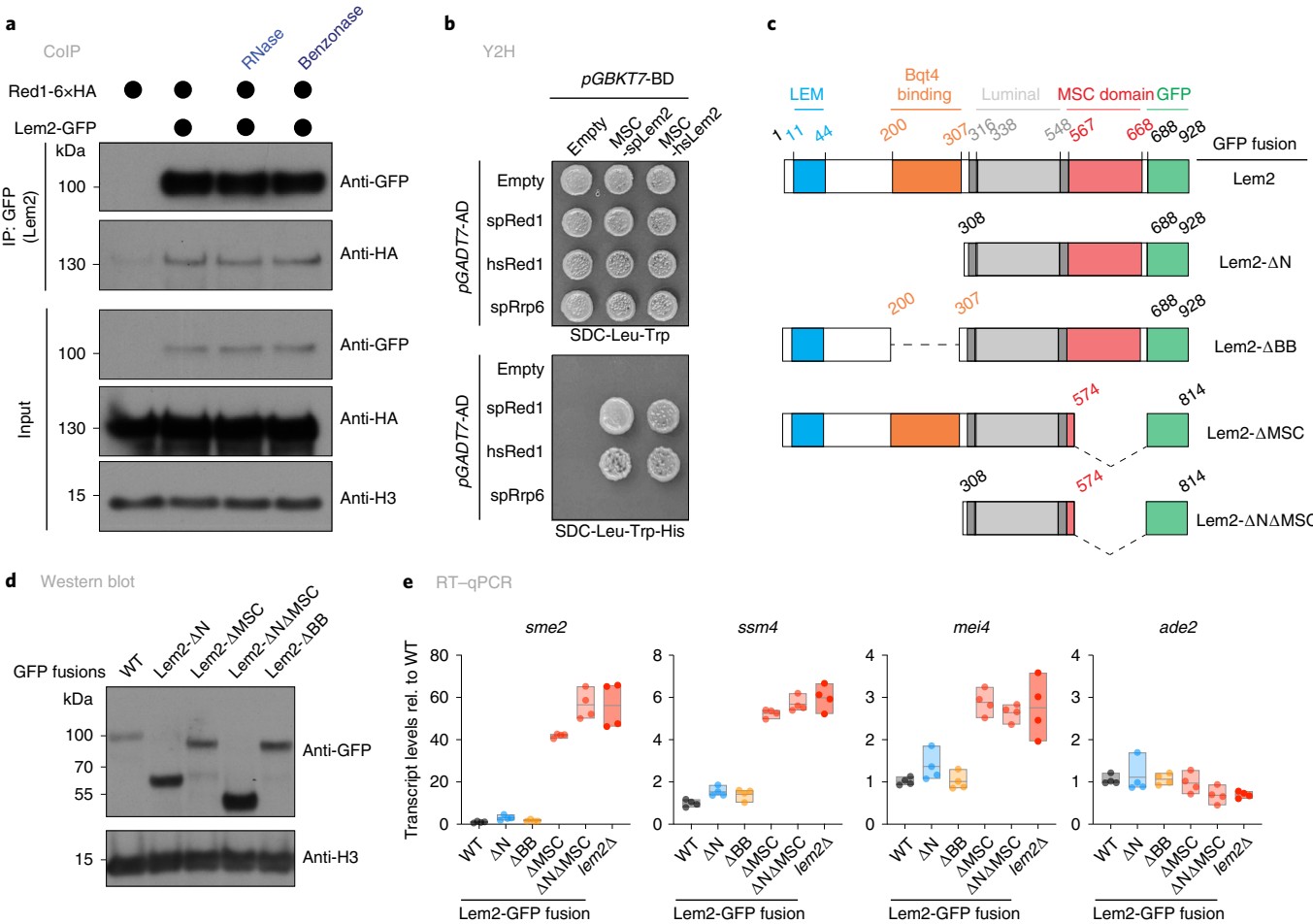

**Fig. 3 | Lem2 physically interacts with the nuclear exosome through the MSC domain. a**, Co-immunoprecipitation of Red1-6xHA with Lem2-GFP in untreated cells, or cells treated with RNase or benzonase. H3 served as loading control. Shown is a representative example of experiments reproduced at least three times. **b**, Y2H analysis of spRed1 (*S. pombe* Red1), hsRed1 (*H. sapiens* homolog of Red1 corresponding to ZFC3H1 492–1308 aa), or spRrp6 with MSC-spLem2 or MSC-hsLem2, grown for 3 days on medium with increasing auxotrophy selection (SDC, synthetic media with dextrose and complete amino acid mix; -Trp, -Leu, -His, without tryptophan, leucine, histidine, respectively). Fusions with Gal4 activation domain (*pGADT7*-AD) or Gal4 DNA-binding domain (*pGBKT7*-BD) are shown. **c**, Schematic representation of Lem2 truncation constructs. Protein domains and positions (amino acids) are highlighted. All constructs were C-terminally GFP-tagged and expressed from the endogenous locus. **d**, Immunoblot of Lem2-GFP constructs in **c**. H3 served as a loading control. **e**, Transcript levels of *sme2*, *ssm4*, *mei4*, and *ade2*, quantified by RT–qPCR on the Lem2 truncation mutants shown in **c**. Data are normalized to *act1* transcript levels and shown relative to WT (*n* = 4 independent biological replicates). The individual replicates are shown in a floating bar plot and the line depicts the median.

results, Lem2-MSC and full-length Red1 interacted in the Y2H assay (Fig. 3b). Lem2-MSC also associated with Iss10, but not with Rrp6, Pab2, or Mtl1 (Extended Data Fig. 4b). In humans, the helicase Mtr4 associates with the zinc finger protein ZFC3H1, forming the PAXT complex, the human homolog of MTREC[12]. Remarkably, we found a strong interaction between hLEM2-MSC and ZFC3H1 across species, indicating that binding between Lem2 and MTREC is conserved (Fig. 3b).

Next, we analyzed selected targets by RT–qPCR to test whether the MSC domain mediates the repression of exosome substrates using a series of published Lem2 truncation mutants (Fig. 3c)[42]. We confirmed the expression of these constructs by immunoblots (Fig. 3d). N-terminal truncations of Lem2 did not impact the level of *sme2* or other transcripts. However, mutants lacking the C-terminal region largely phenocopied the repression defect of the full deletion (Fig. 3e). To further test whether the perinuclear localization of the MSC domain is important for target repression, we generated a soluble C-terminal Lem2 fragment lacking the two transmembrane domains. This fragment was fused to GFP and the SV40 nuclear

localization signal (NLS), and expressed using either the endogenous *lem2* promoter or the strong *TEF* promoter (Extended Data Fig. 4c). Both constructs produced a diffuse nuclear pattern that differs from the rim shape of full-length Lem2 (Extended Data Fig. 4d). Notably, expression of soluble MSC-GFP failed to suppress the accumulation of meiotic transcripts in *lem2Δ* cells regardless of their protein levels (Extended Data Fig. 4e,f), similar to previous observations of heterochromatin transcripts[38]. We therefore conclude that Lem2 cooperates with the nuclear exosome through Red1 interaction and that perinuclear localization of the MSC domain is crucial for regulating exosome targets.

**Lem2 regulates silencing of exosome targets at the nuclear periphery.** Our data (Extended Data Fig. 4f) and previous studies[38,43] indicate that perinuclear location of Lem2 is critical for its gene repression function. This implies that exosome targeting or degradation of Lem2-dependent RNA substrates takes place at the nuclear periphery. To test this hypothesis, we first examined the subnuclear localization of the *sme2* RNA, which contains 25 DSR

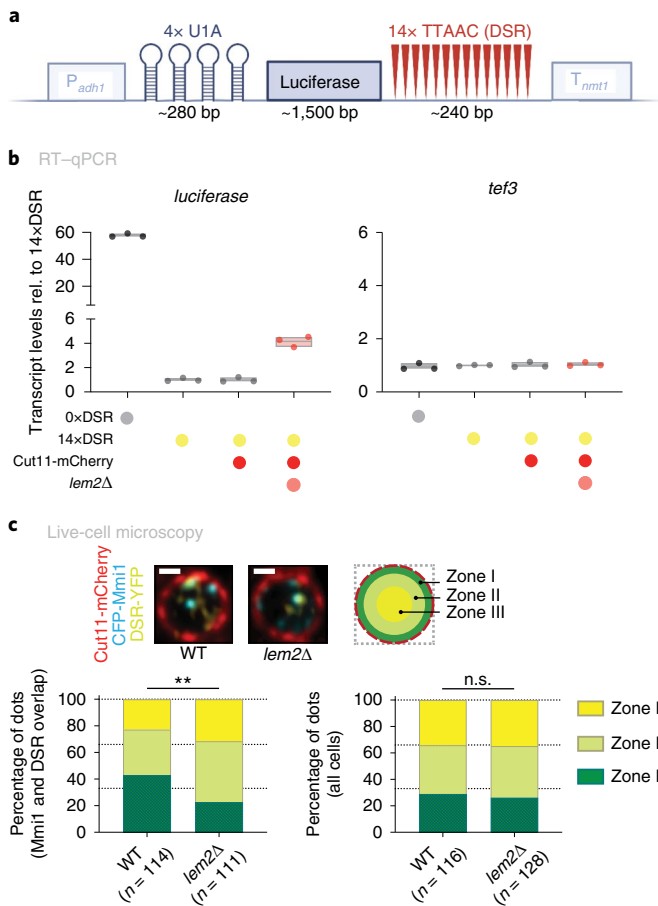

**Fig. 4 | Exosome substrates localize at the nuclear periphery. a**, Schematic representation of the engineered DSR-containing construct, adapted from ref. [23]. Luciferase is expressed with 14 copies of DSR and 4 copies of the U1A tag. $P_{adh1}$, $adh1^+$ promoter; $T_{nmt1}$, $nmt1^+$ terminator. **b**, Transcript levels of *luciferase* and *tef3*, quantified by RT–qPCR in strains encoding 0×DSR copies (0×DSR), 14×DSR copies (14×DSR), or 14×DSR and the NE marker in a WT (14×DSR Cut11-mCherry) or *lem2Δ* background (14×DSR Cut11-mCherry *lem2Δ*). Data are normalized to *act1* transcript levels and shown relative to the 14×DSR strain ($n = 3$ independent biological replicates). The individual replicates are shown in a floating bar plot, and the line depicts the median. Coloured dots underneath the x-axis denote combinations of conditions shown left of the graph. **c**, Top left, representative images from live-cell microscopy of the DSR-containing strain, with CFP-Mmi1 and Cut11-mCherry in a WT or *lem2Δ* background. Cut11-mCherry marks the NE. A single z-stack is shown. Top right, schematic representation of the *S. pombe* nucleus, divided into three equal areas (zones) designated I–III. Bottom, quantification of CFP-Mmi1 location in WT and *lem2Δ* backgrounds relative to the periphery expressed in percentage of dots. Locations of CFP-Mmi1 foci were determined in cells in which the Mmi1 dot exclusively overlapped with the DSR dot (left) and in all cells (right); *n* denotes the number of cells counted in two independent experiments. Statistical analysis was performed using $\chi^2$ test. **$P = 0.0048$; n.s., not significant ($P = 0.7469$). Scale bar, 1 μm.

motifs and is one of the most upregulated transcripts in the *lem2Δ* mutant (Fig. 1a,b). Using single-molecule fluorescence in situ hybridization (smFISH), *sme2* transcripts were readily detectable in vegetative cells lacking Red1 or in wild-type (WT) cells undergoing meiosis (Extended Data Fig. 5a), during which *sme2* forms the Mei2 dot[31]. However, *sme2* smFISH signals were undetectable in WT or *lem2Δ* cells during mitotic growth, precluding further

analysis (Extended Data Fig. 5a). We overcame this technical challenge using an engineered strain[23], which expresses a reporter containing 14 DSRs and 4 U1A small nuclear RNA (snRNA) stem loops that can be visualized by co-expressing U1A-yellow fluorescent protein (YFP) (Fig. 4a). We confirmed that this reporter undergoes Iss10-, Red1-, and Lem2-dependent transcript elimination (Fig. 4b and Extended Data Fig. 5b). Using live-cell imaging, we studied the localization of the DSR-containing RNA reporter relative to the nuclear periphery (Cut11-mCherry). In both WT and *lem2Δ* cells, expression of the 14×DSR reporter resulted mostly in a single dot. We next determined the frequency at which the DSR dot localizes to specific nuclear areas using a zoning assay (Extended Data Fig. 5c)[53]. Interestingly, the DSR dot preferentially localized to the nuclear periphery in 40% of WT cells (zone I), whereas only 20% of the *lem2Δ* cells showed this pattern, indicating that Lem2 promotes the perinuclear localization of this exosome RNA substrate (Extended Data Fig. 5c). This result differed from the localization of genomic loci, as shown for the lacO array/GFP-lacI marked *sme2⁺* locus, which did not preferentially appear at the nuclear periphery (Extended Data Fig. 5d). DSR-containing transcripts localize within nuclear Mmi1 foci, which have been proposed to be degradation sites of meiotic transcripts[13,18,26]. We therefore assessed the subnuclear localization of Mmi1 and the DSR reporter. Expression of CFP-Mmi1 resulted in several foci, some of which overlapped with the DRS reporter dot, as previously reported[23]. Notably, foci that co-localized with the DSR dot displayed Lem2-dependent perinuclear localization (43% versus 23% for zone I; Fig. 4c), implying this exosome substrate is recognized close to the NE. In contrast, the total pool of CFP-Mmi1 foci showed a general subnuclear distribution in WT and *lem2Δ* strains (29% versus 26% for zone I) (Fig. 4c). These data indicate that Lem2 regulation of exosome targets occurs at the nuclear periphery, where a subfraction of the nuclear pool of the elimination factor Mmi1 localizes.

**Lem2 assists exosome substrate targeting.** Given its interaction with Red1 and role in repression and localization of exosome targets (Figs. 2–4), we speculated that Lem2 supports RNA turnover through recognition and handover to MTREC. The MSC domain of the human Lem2 homolog MAN1 (hMAN1) binds to DNA in vitro[54] and contains an RNA-recognition motif[55]. However, *S. pombe* homologs lack this C-terminal extension, and whether Lem2 can bind RNA is unknown.

We used RNA immunoprecipitation (RIP) followed by RT–qPCR to assess binding of *sme2* or *ssm4* transcripts, which contain several DSRs (Fig. 5a)[3]. When immunoprecipitating Lem2-GFP, we were unable to detect those transcripts (Fig. 5b). We therefore tested the alternative hypothesis that Lem2 plays an accessory role in loading RNAs onto exosome-targeting factors. We expressed GFP-Mmi1 and Red1-Myc from their endogenous loci and confirmed that the epitope-tagged versions are functional (Extended Data Fig. 6a). In agreement with previous reports[2,13], we found that *sme2* and *ssm4* transcripts were abundantly enriched with Mmi1 and Red1 (Fig. 5c,d). Strikingly, deletion of *lem2⁺* markedly reduced binding of these transcripts to Mmi1 and Red1 (Fig. 5c,d). Binding of exosome substrates to Mmi1 was less affected in the absence of Red1 (Extended Data Fig. 6b), which is proposed to act downstream of substrate recognition by Mmi1 (ref. [4]). Moreover, these substrates can be captured by Mmi1 even when they accumulate to high levels, as seen in *red1Δ* cells, implying that the substrate-binding capacity of Mmi1 is not limited under these conditions (Extended Data Fig. 6c). Together, these results reveal a critical role for Lem2 in the early step of RNA recognition.

**Multiple pathways contribute to exosome-mediated RNA degradation.** During vegetative growth, various exosome factors assemble into single or multiple nuclear foci, which may be sites

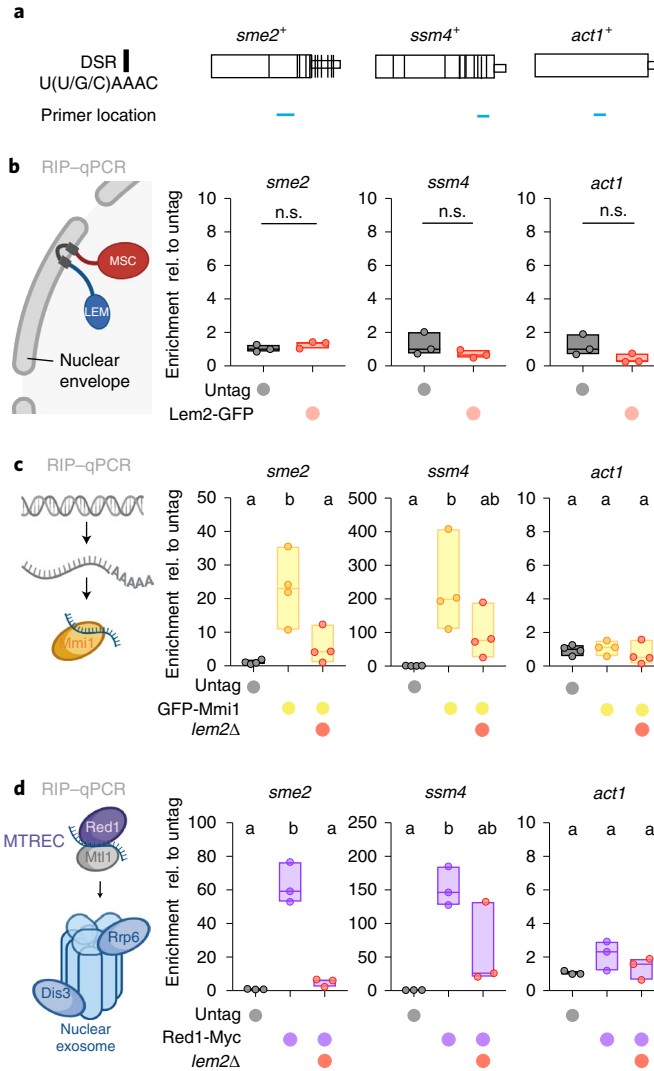

**Fig. 5 | Lem2 promotes binding of RNA targets with exosome-targeting factors. a**, Schematic representation of DSR-containing genes *sme2+* and *ssm4+*, and control gene actin (*act1+*). Sizes were adjusted to match lengths. DSR location was adapted from ref. [51]. DSR location (vertical black lines) and qPCR primer location (horizontal blue lines) are indicated. **b–d**, Transcript binding to Lem2-GFP (**b**), GFP-Mmi1 (**c**), and Red1-Myc (**d**), assayed by RIP–qPCR analysis. Data from independent biological replicates are divided by the input and shown relative to the median of the untagged strain (for **b** and **d**, $n = 3$; for **c**, $n = 4$). The individual replicates are shown in a floating bar plot and the line depicts the median. For **b**, statistical analysis was performed using two-tailed Student's *t*-test; n.s., not significant. For **c** and **d**, statistical analysis was performed using one-way ANOVA, and letters denote groups with significant differences as determined by Tukey's post hoc tests at $P < 0.05$.

of RNA degradation[13,24,28,56]. While EMC and MTREC can form independent foci, their mutual interaction depends on Iss10, which physically interacts with Red1 (refs. [9,26]). We investigated whether the formation and localization of these nuclear foci require Lem2. Using live-cell imaging, we confirmed that the exosome-targeting factors Mmi1, Erh1, and Red1 form multiple nuclear foci. However, in agreement with our findings for Mmi1 (Fig. 4c), we observed neither a preferential perinuclear enrichment of these foci in WT cells nor altered formation or localization upon deletion of *lem2+* (Extended Data Fig. 7a). Moreover, although Iss10 is critical

for bridging the interaction between Red1 and Mmi1 (refs. [9,11,13,26]), we found that GFP-Mmi1 association with Red1-6×HA was unaffected by *lem2+* deletion (Extended Data Fig. 7b).

Since *erh1Δ* and *iss10Δ* mutants also showed little functional overlap with the *lem2Δ* mutant (Fig. 2), we tested whether RNA degradation involving Lem2 is mediated through an Erh1- and Iss10-independent pathway. When examining the *erh1Δ lem2Δ* double mutant, we found a synthetic increase of several meiotic transcripts, but not of snoRNAs, which are processed independently of the Mmi1 elimination pathway (Fig. 6a). Interestingly, although *mei4* and *ssm4* transcripts were also increased in the *lem2Δ iss10Δ* double mutant, this was different for *sme2* and *mei3*, whose transcript levels were unaltered in the single *iss10Δ* mutant and not additionally increased in the *lem2Δ iss10Δ* double mutant (Fig. 6b). Indeed, when studying upregulated transcripts genome-wide, we found complementary patterns in *lem2Δ* and *iss10Δ*, but rarely an additive increase in the *iss10Δlem2Δ* double mutant (Fig. 6c). This suggests that Lem2 and Iss10 act largely through mutually exclusive pathways, whereas Lem2 appears to have overlapping functions with other factors, such as TRAMP (Extended Data Fig. 7c). Because Mei2 dot assembly during early meiosis causes inactivation of Mmi1-dependent elimination[2], we also tested whether the accumulation of exosome targets in *lem2Δ* cells was indirectly caused by stabilizing *sme2*. However, deleting *sme2+* did not suppress the accumulation of meiotic transcripts or snoRNAs in *lem2Δ* cells (Extended Data Fig. 7d), further arguing that Lem2 plays a direct role in their degradation. Together, these results imply that RNA degradation is coordinated through distinct degradation pathways that depend on Lem2 and Iss10 and differ in their substrate specificity.

Previously, Lem2-dependent heterochromatin assembly was reported to be regulated by nutrient availability[43]. Heterochromatic transcripts are no longer repressed in a Lem2-dependent manner when growth conditions are restricted using EMM (Edinburgh minimal medium; Extended Data Fig. 7e). Since nitrogen starvation promotes sexual differentiation and entry into meiosis, we tested whether EMM lacking nitrogen (EMM-N) impacts Lem2 functions. Intriguingly, we found that *sme2* transcript levels increased in WT cells in EMM and EMM-N, similar to what is seen in *lem2Δ* cells, suggesting that Lem2 becomes inactivated under these growth conditions (Fig. 6d). To further investigate whether Lem2 activity is regulated during meiotic onset, we grew mating-competent h[90] cells in rich medium and analyzed *sme2* transcripts upon transfer to EMM-N in a time-course experiment (Fig. 6e). *sme2* transcripts steadily increased in WT cells upon nitrogen starvation, whereas this transcript was already upregulated in *lem2Δ* cells in rich medium and did not further increase upon starvation (Fig. 6e). Although these meiotic transcripts did not accumulate in WT cells to the same level as in *lem2Δ* cells, it is possible that some cells within the population may not have entered the meiotic program, likely explaining the weaker phenotype in WT cells. On the basis of these results, we propose that nutrient-dependent inactivation of Lem2 contributes to the fine-tuning of meiotic transcript accumulation during early meiosis. Beyond this role in meiosis, Lem2 is key to a spatially and functionally distinct pathway that operates in post-transcriptional regulation at the NE.

## Discussion

Transcriptionally silent chromatin often localizes to the nuclear periphery, which is thought to provide a specialized compartment for gene repression[57]. However, whether the NE influences broader modes of gene regulation is largely unknown. Here, we demonstrate that the conserved INM protein Lem2 collaborates with the nuclear exosome to control and process ncRNAs and meiotic transcripts. Several lines of evidence argue that this Lem2-mediated regulation occurs post-transcriptionally and is distinct from its previously described function in heterochromatin silencing[38,43]:

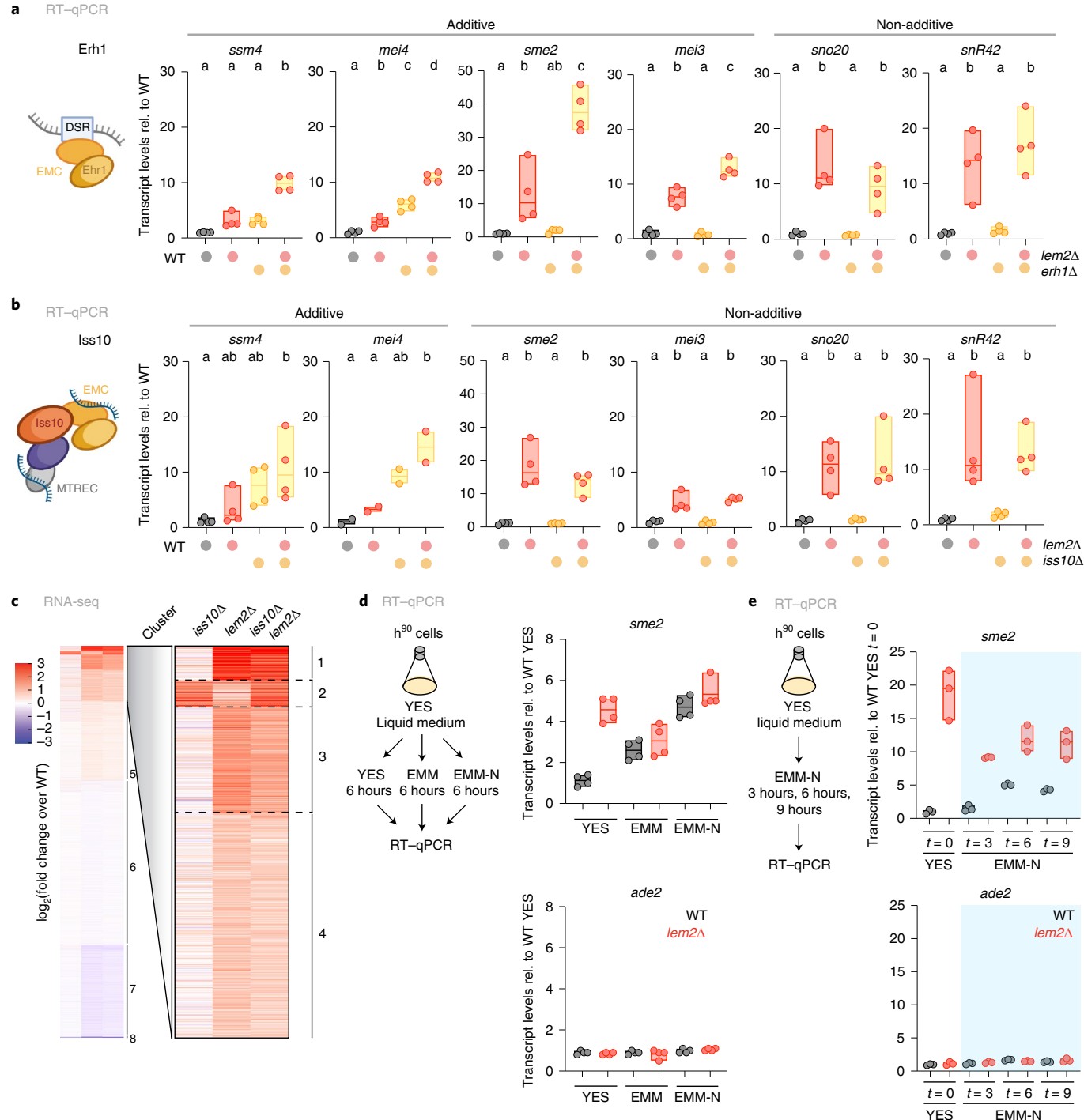

**Fig. 6 | RNA regulation by Lem2 at the nuclear periphery occurs independently of exosome factors associated with nuclear foci. a,b,** Left, scheme of the subunit that was mutated on its own or together with Lem2. Right, transcript levels of *ssm4*, *mei4*, *sme2*, *mei3*, *sno20*, and *snR42*, quantified by RT–qPCR in the indicated strains (**a**, *erh1Δ*; **b**, *iss10Δ*). Substrates controlled in an additive or non-additive manner are indicated above. Data normalized to *act1* transcript levels are shown relative to WT ($n = 4$ independent biological replicates, except *mei4* transcripts in combination with *iss10Δ*, $n = 2$). The individual replicates are shown in a floating bar plot, and the line depicts the median. Statistical analysis was performed using one-way ANOVA, and letters denote groups with significant differences from Tukey's post hoc tests at $P < 0.05$. **c,** *K*-means clustering of RNA-seq data. Genes were grouped using *K*-means clustering on the basis of their differential expression in the indicated mutants. Eight clusters were generated, and the clusters with most upregulated transcripts in *lem2Δ* (clusters 1–4) were analyzed with AnGeLi with a two-tailed Fisher's exact test and false-discovery rate (FDR) = 0.05 (ref. [46]). The red-blue color scale represents the log$_2$(fold change expression) relative to WT. **d,** Left, scheme showing the experimental setup and medium shift. Right, transcript levels of *sme2* and *ade2* quantified by RT–qPCR in the indicated strains (gray, WT; red, *lem2Δ*) and medium (YES or EMM-N). **e,** Left, scheme showing the experimental setup and media shift. Right, transcript levels of *sme2* and *ade2*, quantified by RT–qPCR in the indicated strains (gray, WT; red, *lem2Δ*), time points ($t = 0$, 3, 6 or 9 hours) and medium (YES or EMM-N). For **d** and **e**, data from independent biological replicates are normalized to euchromatin expression and shown relative to the WT strain in YES (**d**; $n = 4$) or the WT strain at $t = 0$ in YES (**e**; $n = 3$).

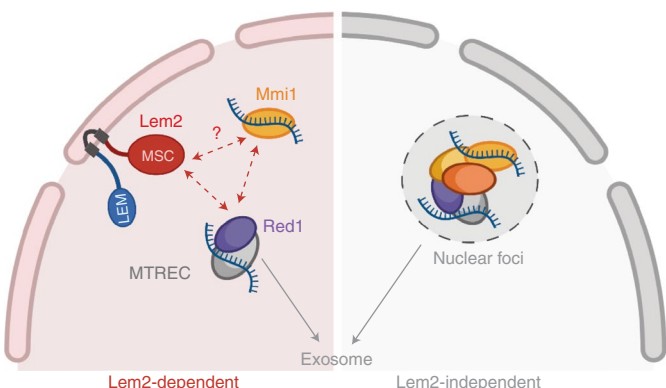

**Fig. 7 | Working model for Lem2 cooperation with the nuclear exosome for substrate degradation at the nuclear periphery.** Left, Lem2 cooperates with the nuclear exosome machinery at the nuclear periphery by directly interacting with Red1 and assisting in substrate binding by Mmi1 and MTREC. Absence or inactivation of Lem2 results in impaired DSR substrate location at the periphery, reduced binding to exosome-targeting factors, and eventually an increase in untimely transcript expression due to lower RNA degradation (Lem2-dependent). Lem2 may also interact with other factors associated with Mmi1. Right, nuclear foci are formed by multiple exosome factors in an Iss10-dependent manner and likely represent additional sites of RNA degradation (Lem2-independent). Both pathways show partial redundancy for most meiotic transcripts, whereas ncRNAs are mainly degraded by the Lem2-dependent pathway.

(1) although *lem2+* deletion causes accumulation of various targets (ncRNA, LTRs, and heterochromatic transcripts), Pol II abundance is increased at heterochromatin but not at ncRNA genes; (2) repression of targets co-regulated by Lem2 and the exosome is unaffected by silencing factors (Clr4/HP1, RNAi, SHREC) that work with Lem2 at heterochromatin; (3) mutants lacking Lem2 and exosome factors (Rrp6, MTREC) display overlapping but non-additive phenotypes in the repression of several meiotic transcripts; (4) Lem2 physically interacts with the MTREC subunit Red1 via its MSC domain; (5) Lem2 does not affect chromatin binding of Mmi1 and Red1; (6) instead, Lem2 promotes Mmi1 and Red1 binding to RNA substrates; (7) Lem2 is critical for the localization and association of a DRS-containing transcript with a subnuclear pool of Mmi1 at the NE. On the basis of these findings, we propose that Lem2 recruits exosome co-factors to the nuclear periphery to coordinate post-transcriptional RNA processing and degradation, a function independent of heterochromatin silencing.

Despite major differences in transcriptional and post-transcriptional regulation, Lem2 plays a general role in increasing the local concentration of factors involved in these processes at the nuclear periphery. As previously shown, Lem2 promotes the localization of the repressor complex SHREC to heterochromatin[38]. Similarly, we find that Lem2 facilitates the association of MTREC with RNA substrates (Fig. 5) and interacts with Red1 through its MSC domain (Fig. 3). Lem2 therefore employs a common mechanism for interaction with its partners, which is in agreement with the MSC-dependent recruitment of members of the ESCRT pathways in NE repair and other functions reported for Lem2 homologs[58–62]. Thus, Lem2 may provide a general INM recruitment platform for interaction with various partners. How Lem2 specifically coordinates these different functions remains unknown.

Various membrane-less nuclear bodies have been assigned to specific functions, such as the Cajal body involved in snRNA and snoRNA modification and assembly[63]. RNA turnover has also been proposed to occur within subnuclear foci formed by exosome factors in an Iss10-dependent manner[13,24,26,28]. This raises the question of how these nuclear foci relate to Lem2-mediated regulation. Although an engineered exosome substrate localizes more frequently at the nuclear periphery (Fig. 4), such preferential localization was not seen for Red1, Erh1, or Mmi1 foci (Extended Data Fig. 7). Nonetheless, we found that a subpopulation of Mmi1 foci co-localizing with this substrate also localized to the NE in a Lem2-dependent manner (Fig. 4). Interestingly, Mmi1 and another nuclear RNA-processing complex, CCR4–NOT, have been shown to associate with the nuclear rim protein Amo1 (ref. [39]), implying that multiple spatially distinct degradation pathways exist. Indeed, *lem2Δ* mutants that also lack the EMC or TRAMP complexes show synergistic upregulation of various transcripts (Fig. 6). This likely explains why the phenotype is weaker than that of *red1Δ* and *rrp6Δ* mutants (see model, Fig. 7). The impact on distinct pathways would explain why Iss10 is critical for nuclear foci formation while having only a minor role in meiotic transcript turnover in vegetative cells[10,26,28,29]. We speculate that separate pathways allow differential regulation and distinct substrate specificity. We further observe that Lem2 mediates some MTREC-independent function in snoRNA processing (Fig. 2). Further work should elucidate Lem2's specific role in the broad spectrum of its substrates.

The expression of meiotic genes is toxic during vegetative growth and therefore tightly regulated[64]. Under nutrient starvation, cells mate and undergo meiosis, which requires precise orchestration of the meiotic gene expression program[65]. Nitrogen starvation is signaled by TOR pathway inactivation, resulting in the dephosphorylation and degradation of Iss10 (ref. [29]). Similarly, cells shifted from rich to minimal medium show upregulation of key meiotic regulators in a Lem2-dependent manner (Fig. 6e). Hence, we propose that Lem2 is part of a regulatory circuit that fine-tunes gene expression in response to environmental cues. How different growth conditions alter Lem2 activity remains unknown. Lem2 transcript (Extended Data Fig. 7f) and protein levels[43] are unaltered under minimal growth conditions, suggesting that Lem2 may undergo post-translational modifications that impact association with the NE or downstream factors. Thus, it will be interesting to address how nutritional cues affect Lem2 function and to elucidate the underlying signaling cascade.

Our data demonstrate that RNA degradation is not a generic process, but a spatially specific mode of regulation that is critical in the biological response to environmental changes. Since both Lem2 and exosome-targeting complexes are found in higher eukaryotes, we propose that this pathway is broadly conserved. Indeed, the nuclear exosome localizes to the NE in other organisms, including *Drosophila*[66]. Moreover, Lem2 binding to MTREC is conserved for its human homolog, the PAXT complex (Fig. 3b). Further studies examining the nuclear location of the exosome in fission yeast and higher eukaryotes may shed light on the mechanisms that collaborate to regulate this complex machinery.

## Online content

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

## Methods

**Yeast techniques, plasmids, and strains.** A list of the strains used in this study can be found in Supplementary Tables 1 and 2. All plasmids used in this study are listed in Supplementary Table 3.

Standard medium and genome engineering methods were used. Cells were grown in rich medium (YE5S aka YES) except for the data shown in Figures 3d and 6a and Extended Data Figure 6e,f (cells were initially grown in minimal medium (EMM) then shifted for 12 hours into YES) and Figure 6c (cells were initially grown in YES then shifted into EMM-N). Strains expressing constructs derived from *pREP81* vectors (shown in Extended Data Fig. 3a) were grown in EMM-leu.

Strains expressing epitope-tagged proteins were generated through homologous recombination using *pFA6a* or *pYM*-based vectors[67] and expressed from the chromosomal locus using the endogenous promoter. For Lem2-GFP fusions expressing strains, the respective fragments were cloned into a *pJK210* vector and integrated into *lem2Δ* background strains in the endogenous locus. The soluble MSC-GFP overexpression construct was expressed under control of the *TEF* (*Ashbya gossypii*) promoter.

**RNA-seq library preparation and data analysis.** For RNA-seq, 1 µg of RNA was used as starting material to prepare libraries, following the manufacturer's instructions for NEBNext Ultra Directional RNA Library Prep Kit for Illumina (NEB). Two or more biological replicates were used for generating libraries in parallel. For *man1Δ* and *clr4Δ*, three biological replicates were processed. Single-end, 50-bp sequencing of libraries was performed on a HiSeq1500 sequencer in the LAFUGA core facility at the Gene Center, LMU Munich. Raw reads were de-multiplexed using Je (v1.2). Adapter-trimmed reads were aligned to the *S. pombe* reference genome (ASM294v2.27) and a custom GTF file using STAR (v2.7.3a), then processed using RSEM (v1.3.3). Differential expression was analyzed using the DESeq2 (v1.22.2) and tximport (v1.10.1) R libraries. For PCA plotting, data were batch normalized using the sva (v3.30.1) R library. Bedgraph coverage files for plus and minus strands were generated using genomecov (bedtools v2.29.1). RNA-seq data have been deposited in Gene Expression Omnibus (GEO) database under the accession number GSE174347. The full code for all NGS-related workflow is available at: https://github.com/Tsvanemden/Martin_Caballero_et_al_2021.

**RT–qPCR analyses.** RT–qPCR experiments were performed as previously described[68]. Briefly, cells were lysed by bead beating (Precellys 24, Bertin instruments) using TRIzol reagent and zirconia/silica beads (BioSpec), followed by centrifugation at 13,000 r.p.m. for 15 minutes at 4 °C. Recovered supernatant was extracted with chloroform and reprecipitated with isopropyl alcohol. Resuspended RNA was treated with DNaseI, and 10 mg of RNA was used in standard RT reactions using oligo((dT)20-N) primers. cDNAs were quantified by qPCR using primaQUANT SYBR Master mix (Steinbrenner Laborsysteme) and a QuantStudio 3 or QuantStudio 5 Real-Time PCR system (Applied Biosystems/Thermo Fisher). Data from 2–6 independent biological replicates are shown as individual data points together with the median.

For northern blotting and RT–qPCR analysis of snoRNAs (Fig. 2), total RNAs were prepared using the hot acid phenol method and treated with DNAse (Ambion). Concentrations were measured with a Nanodrop. Two micrograms of DNAse-treated RNAs were denatured at 65 °C for 5 minutes in the presence of strand-specific primers. Reactions were carried out with 100 units Maxima H minus Reverse Transcriptase (Thermo Fisher Scientific) at 50 °C for 30 minutes. The enzyme was then denatured at 85 °C for 5 minutes, and reactions were diluted to 1:10 ratio. Experiments included negative controls without Reverse Transcriptase. Samples were analyzed by qPCR with the SYBR Green Master Mix (Roche) and a LightCycler LC480 apparatus (Roche). Quantification was performed using the ΔCt method.

Primers used for qPCR are listed in Supplementary Table 4. Expression values for WT and mutants were calculated by normalizing to *act1* and then dividing by the mean of all samples from the same experiment (group normalization), as previously described[69]. When analyzing single and double mutants for epistatic interactions, statistical testing was performed using R. Multiple testing was performed using ANOVA followed by a Tukey's post hoc test at a 0.05 significance level.

**Northern blotting.** Three micrograms of RNA were separated on a 2% agarose gel and transferred overnight by capillarity on a nylon membrane (GE Healthcare) in SSC 10× buffer. RNAs were then UV-crosslinked to the membrane using a Stratalinker apparatus. In vitro-transcribed dig-labeled RNA probes were generated and incubated with the membrane using the Dig-Northern-Starter Kit (Roche), following the manufacturer's instructions. Membranes were washed twice in 2× SSC 0.1% SDS and once in 1× SSC 0.1% SDS, for 10 minutes at 65 °C. Revelation was done according to the kit instructions using Chemidoc Imaging MP detection device (Biorad). Oligonucleotides used to generate DNA templates for RNA probes are listed in Supplementary Table 4.

**Y2H assays.** Constructs were cloned into either *pGADT7* or *pGBKT7* vectors (Clontech). The *S. cerevisiae* strain Y2H Gold (Takara) was used to co-transform the plasmids, following the manufacturer's instructions. Spotting assays were performed 3–5 days after transformation of the plasmids. Different dropout mixes were used to assess the strength of the interaction: SDC-Leu-Trp (Formedium), SDC-Leu-Trp-His (Formedium) and SDC-Leu-Trp-His-Ade (Formedium).

**Co-immunoprecipitation assays.** CoIP assays were performed following a previously described protocol[70] with a few modifications in the lysis buffer. The lysis buffer contained 50 mM HEPES pH 7.4, 100 mM NaCl, 10% glycerol, 1 mM EDTA pH 8, 2.5 mM $MgCl_2$, 0.5% NP-40, 1× complete EDTA-free protease inhibitor cocktail (Roche), 2 mM PMSF (Serva), 20 mM *N*-ethylmaleimide (NEM, Sigma). Cell lysates were prepared by resuspending the pellets from 150 to 200 $OD_{600}$ units in 800 µl lysis buffer. Cells were lysed by bead beating (Precellys 24, Bertin instruments) with zirconia/silica beads (BioSpec), and lysates were cleared by centrifugation (800*g*, 5 min). Clarified extracts were incubated with pre-equilibrated GFP-Trap or Myc-Trap (Chromotek) for 1.5 hours at 4 °C. Following immunoprecipitation, the bound material was incubated with RNase (Roche) or benzonase (Sigma) in the reactions, as indicated. Beads were washed four times with lysis buffer and two times with wash buffer (50 mM Tris pH 7.5, 100 mM NaCl, 1 mM EDTA pH 8, 2.5 mM $MgCl_2$). Proteins were eluted with 30 µl HU buffer (8 M urea, 5% SDS, 200 mM Tris-HCl pH 6.8, 20 mM dithiothreitol (DTT) and bromophenol blue 1.5 mM) and analyzed by immunoblotting.

**Immunoblotting.** Cells corresponding to $OD_{600} = 1$ ($\approx 2 \times 10^7$ cells) were pelleted from a suspension culture grown to mid-log phase. Total protein extracts were made using trichloroacetic acid (TCA, Sigma) precipitation[68]. Proteins were solubilized in HU buffer (8 M urea, 5% SDS, 200 mM Tris-HCl pH 6.8, 20 mM dithiothreitol (DTT, Sigma) and bromophenol blue 1.5 mM). Proteins were resolved on NuPAGE 4–12% gradient gels (Invitrogen) or self-made 8% gels. Proteins were transferred onto PVDF membranes (polyvinylidene fluoride membranes, GE Healthcare) and analyzed by standard immunoblotting techniques using specific antibodies. H3 antibody was used as a loading control.

**Live-cell microscopy.** Live-cell imaging was essentially performed as described[38]. In brief, cells were grown overnight on rich medium (YES) to the logarithmic phase. Prior to imaging, cells were attached with lectin (Sigma) to glass-bottom dishes with a micro well (MatTek). Cells were imaged on a Zeiss AxioObserver Z1 confocal spinning disk microscope with an EMM-CCD camera (Photometrics, Evolve 512) through a Zeiss Alpha Plan/Apo ×100/1.46 oil DIC M27 objective lens. *Z*-stacks were obtained at focus intervals of 0.4 µm. FiJi/ImageJ software was used to measure the distances between the foci and the periphery.

For the imaging of cells expressing CFP-Mmi1, the following setup was used: confocal microscopy was performed at the Core Facility Bioimaging of the Biomedical Center (LMU Munich) with an inverted Leica SP8 X WLL microscope, equipped with 405-nm laser, WLL2 laser (470–670 nm), and acusto-optical beam splitter. Images were acquired with a HC PL APO ×93/1.30 GLYC motCORR-STED WHITE objective, and *Z*-stacks were obtained at focus intervals of 0.25 µm. Images were deconvolved using the SVI Huygens suite and FiJi/ ImageJ software was used to measure the distances between the foci and the periphery.

**Single molecular RNA fluorescence in situ hybridization.** smFISH was performed as described in the previous report with slight modification[71]. Cells of WT (H1N2330), *lem2Δ* (H1N2324), and *red1Δ* (H1N2328) were pre-cultured in minimal medium (EMMG5S) for overnight at 30 °C, then transferred into rich medium (YES) and cultured for 16 hours. The cells (~1 × 10^8) were mixed with one-tenth of meiosis-induced PSB1940 (YY548-13C) cells (~1 × 10^7) as a positive control for smFISH, and then fixed with 4% formaldehyde (Polysciences) at 30 °C for 30 minutes. The cells were labeled with Quasar 570-labeled RNA probe set for sme2 (See Supplementary Table 4 in ref. [71] in detail), then mounted in ProLong Glass Antifade Mountant (Thermo Fisher Scientific).

The cells were observed using ×60 PlanApo N OSC oil-immersion objective lens (numerical aperture (NA) = 1.4, Olympus) on the DeltaVision Elite system (GE Healthcare) equipped with pco.edge 4.2 sCMOS camera (PCO). Chromatic shifts were corrected using Chromagnon software (v0.87) using a bleed-through fluorescence image as a reference[72]. The images were denoised by the ND-safir program[73], deconvolved using the built-in SoftWoRx software (v7.0.0), and then projected by maximum-intensity projection. The brightness of the images was adjusted using the Fiji software[74] for better visualization, without changing the gamma settings.

**RIP assays.** Cell lysates prepared from equal amounts of cells (between 100–165 $OD_{600}$) were fixed with 1% formaldehyde (Sigma) for 15 minutes at RT, followed by quenching 5 minutes at RT with 125 mM glycine (Sigma). Cultures were spun down, washed once with 1×PBS and frozen in liquid nitrogen. Cells pellets were resuspended in lysis buffer (250 mM KCl, 1% Triton X-100, 0.1% SDS, 0.1% Na-deoxycholate, 50 mM HEPES pH 7.5, 2 mM EDTA, 2 mM EGTA, 5 mM $MgCl_2$, 0.1% Nonidet P-40, 20% glycerol) and lysed with glass beads (Roth) in a bead beater (Precellys 24, Peqlab). Fragmented material was sonicated (Qsonica Q800R1) for 1 hour with cycles of 30 seconds ON/OFF at 4 °C. The lysate was cleared and used for immunoprecipitation with 15 µl GFP-trap or Myc-trap (Chromotek). Beads were washed with lysis buffer, and bound material was

eluted from beads with elution buffer (50 mM Tris-HCl pH 8.0, 10 mM EDTA, 1% SDS) with 15 minutes of incubation at RT and 15 minutes at 65 °C. RIP samples along with inputs were de-crosslinked at 95 °C for 10 minutes. Samples were then incubated with 40 µg of proteinase K (Sigma) for 4 hours at 37 °C. RNA was recovered with a phenol-chloroform-isoamyl alcohol extraction (Thermo Fisher Scientific) followed by precipitation with sodium acetate, isopropanol, and glycogen (Thermo Fisher Scientific). Precipitated RNA was digested with DNase I (Thermo Fisher Scientific) for 2 hours at 37 °C. Purified RNA was used for reverse transcription following manufacturer's instructions (Superscript III, Thermo Fisher Scientific) and used for qPCR, as described for RT–qPCR samples. The primers used are listed in Supplementary Table 4.

**Pol II-S5P, H3K9me2, and Red1-Myc ChIP–qPCR assays.** ChIP was performed as previously described[38] with minor modifications, as follows: 100 ml of a 0.5 $OD_{600}$ cell suspension was crosslinked with 1% formaldehyde (Roth) and quenched with 125 mM glycine (Sigma). Following lysis and sonication, solubilized chromatin corresponding to approximately $5–6 \times 10^8$, $4 \times 10^8$, and $9 \times 10^8$ cells was immunoprecipitated with antibodies against Pol II-S5P (25 µl supernatant, kindly provided by A. Ladurner), H3K9me2 (2 µl), and Myc-trap (10 µl, Chromotek), respectively.

RT–qPCR experiments were performed as described above, using the primers listed in Supplementary Table 4. The IP values were divided by the input and corrected for variation by normalizing to the mean of three euchromatin loci ($act1^+$, $tef3^+$, $ade2^+$). The data were shown as relative to the untagged strain.

**Antibodies.** Rat monoclonal anti-HA (3F10, 1:1,000) and mouse monoclonal anti-GFP (B-2, 1:1,000) antibodies were purchased from Roche and Santa Cruz Biotechnology, respectively. Mouse monoclonal anti-H3 (1B1-B2, 1:5,000) was obtained from Active Motif. Rabbit polyclonal anti-Myc (ab9106, 1:2,000, Abcam) or mouse monoclonal anti-Pol II-S5P (3E8, generated by the lab of D. Eick) antibodies were kindly provided by M. Spletter and by A. Ladurner, respectively. Mouse monoclonal H3K9me2 ChIP grade (ab1220) was purchased from Abcam. Secondary antibodies fused to HRP were used for detection (goat anti-mouse HRP 1:3,000, BioRad; goat anti-rat HRP 1:3,000, Merck Millipore; goat anti-rabbit HRP 1:3,000, BioRad)

**Statistics and reproducibility.** Representative results of at least two independent experiments were presented in all of the figure panels for all blots. No statistical methods were performed to predetermine the sample size, and the number of biological replicates was based on similar studies for each experiment. Analyses of the variance were performed, and pairwise differences were evaluated with Tukey's post hoc test using R statistical language (R Development Core Team, 2008); different groups are marked with letters at the 0.05 significance level. For all graphs, data the individual replicates are shown in a floating bar plot, and the line depicts the median. $P$ values were generated using two-tailed Student's $t$-tests or chi-square ($\chi^2$) analyses; n.s., $P \geq 0.05$, *$P \leq 0.05$, **$P \leq 0.01$, ***$P \leq 0.001$, ****$P \leq 0.0001$.

**Reporting summary.** Further information on research design is available in the Nature Research Reporting Summary linked to this article.

## Data availability
All sequencing data that support the findings of this study have been deposited in the National Center for Biotechnology Information Gene Expression Omnibus (GEO) and are accessible through the GEO Series accession number GSE174347. Source data are provided with this paper.

## Code availability
Full code for all NGS-related workflow is available at: https://github.com/Tsvanemden/Martin_Caballero_et_al_2021.

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

## Acknowledgements
We thank members of the Braun lab and A. Ladurner, P. Korber, and M. Halic for fruitful discussions during the study. We thank S. Lall (Life Science Editors) for editorial assistance. We further thank A. Ladurner (LMU Munich, Germany), M. Smolle (LMU Munich, Germany), and M. Halic (St. Jude Children's Research Hospital, Memphis, TN, USA) for strains, plasmids and reagents; A. Yamashita; C. Brönner, M. Halic, and M. Murawska for protocols; T. Straub and T. Schauer from the BMC Bioinformatics core facility for training and counseling; G. Timinszky, J. Preisser, A. Thomae, and the BMC Bioimaging core facility for training and counseling in confocal microscopy and data deconvolution; D. Q. Ding and K. Okamasa for helpful discussion for smFISH; S. Stöcker for assistance in library preparation for RNA-seq analysis; G. Schermann for support in antisense transcript analysis. Schemes in figures were created with BioRender.com. S. B. was supported by the German Research Foundation (DFG) through the collaborative research center CRC 1064 (project ID 213249687-SFB 1064) and the research grant BR 3511/4-1. I. S. was supported by the Leibniz program of the DFG (SI586/6-1). Additional support was provided by the Australian Government through the Australian Research Council's Discovery Projects funding scheme awarded to T. F. (project DP190100423); by the Centre National pour la Recherche Scientifique and the Agence Nationale de la Recherche to M. R. (ANR-16-CE12-0031); and by the JSPS KAKENHI grants to Y. Hirano (JP19K06489, JP20H05891), Y. K. (JP19K23725), and Y. Hiraoka (JP18H05533, JP19K22389).

## Author contributions
L. M. C., R. R. B., and S. B. conceived the study. L. M. C., R. R. B., and T. v. E. performed RNA-seq experiments, and T. v. E. generated the RNA-seq analysis pipeline. N. D., M. C., and L. M. C. performed Y2H assays. L. M. C. and V. N. S. S. performed western blots. M. C. and L. M. C. conducted coIP experiments. L. M. C., V. N. S. S., and S. F.-B. performed RT–qPCR experiments. A. N. and M. R. performed snoRNA northern blots and RT–qPCR experiments. Y. Hirano conducted smFISH experiments. All other experiments were performed by L. M. C. All authors analyzed the data. S.B., Y. Hiraoka, I.S., T.F. and M.R. supervised experiments. L. M. C. and S. B. conceived and wrote the manuscript. Y. Hirano, Y. K., and Y. Hiraoka made original observations confirming independently several key findings. All authors contributed to editing.

## Funding

## Competing interests
The authors declare no competing interests.

## Additional information
**Extended data** is available for this paper at https://doi.org/10.1038/s41594-022-00831-6.

**Correspondence and requests for materials** should be addressed to Sigurd Braun.

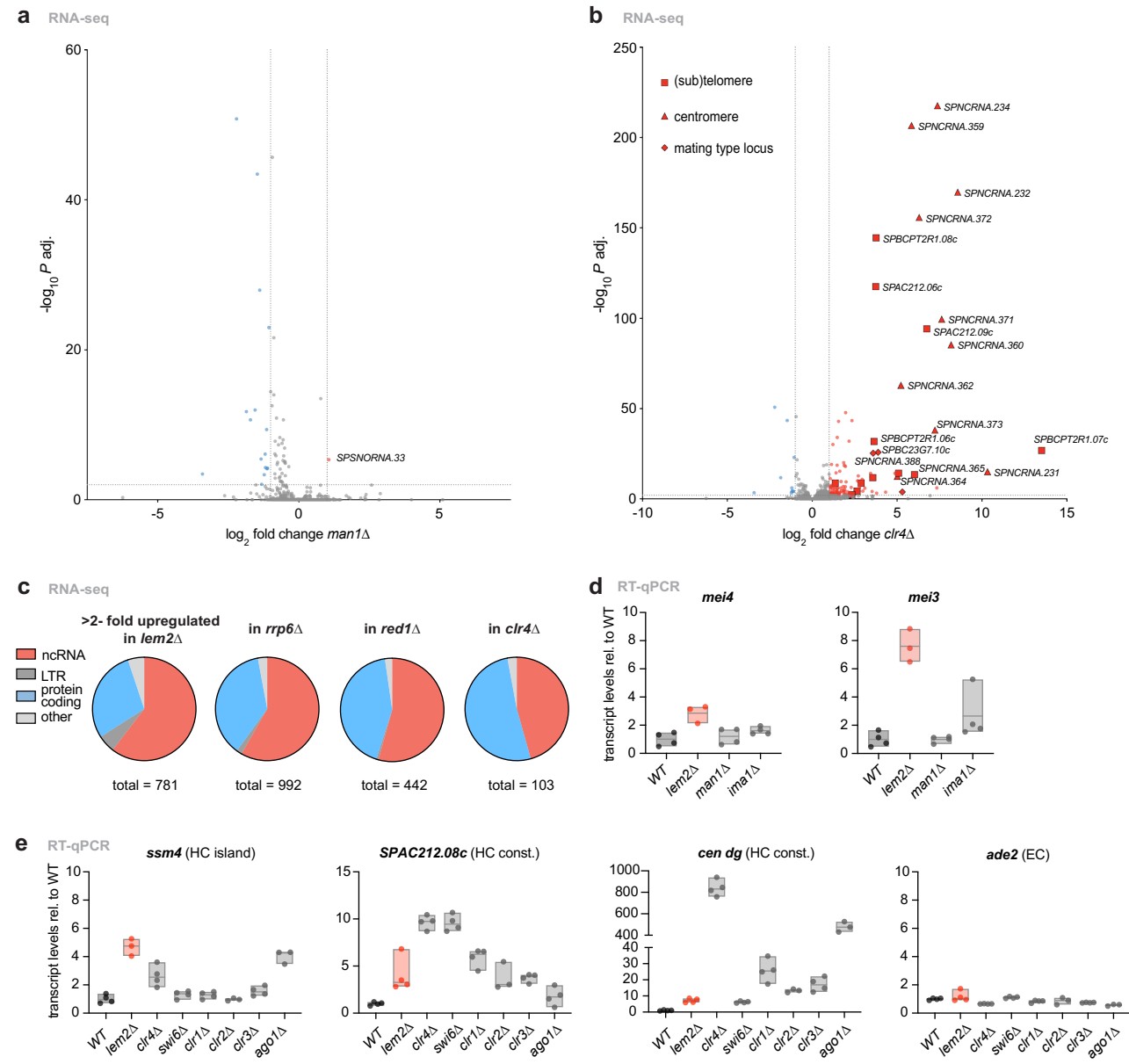

**Extended Data Fig. 1 | Lem2 represses non-coding RNAs and meiotic genes. a**, **b**, Volcano plots depicting statistical significance (Y axis) against fold change (X axis) from the RNA-seq data of *man1Δ* vs WT (**a**) and *clr4Δ* vs WT (**b**). Genes significantly up- (red) or downregulated (blue) are highlighted (log₂ fold change > 1 or < -1 with *P* adj. value < 0.01 by the Wald test, as implemented within the DESeq2 framework). **c**, Pie charts showing of ncRNA, LTR, protein coding, other transcripts (pseudogene, rRNA, snoRNA, snRNA, tRNA). Left: genome-wide distribution of transcript features in a WT genome. Right: transcript feature distribution of the significantly upregulated transcripts in the indicated mutants (log₂ fold change > 1 with *P* adj. value < 0.01 by the Wald test). **d**, Transcript levels of *mei4* and *mei3* analyzed by RT-qPCR qPCR (*n* = 4 independent biological replicates; except *lem2Δ*: *n* = 3). **e**, Transcript levels of *ssm4*, *SPAC212.08c*, *cen dg*, and *ade2* analyzed by RT-qPCR (*n* = 4 independent biological replicates; except *clr2Δ*, *ago1Δ*: *n* = 3). For (**d, e**), data are normalized to *act1* transcript levels or the average of selected euchromatic genes (*act1⁺*, *tef3⁺*, *ade2⁺*), respectively, and shown relative to WT. The individual replicates are shown in a floating bar plot and the line depicts the median. HC, heterochromatin; EC, euchromatin.

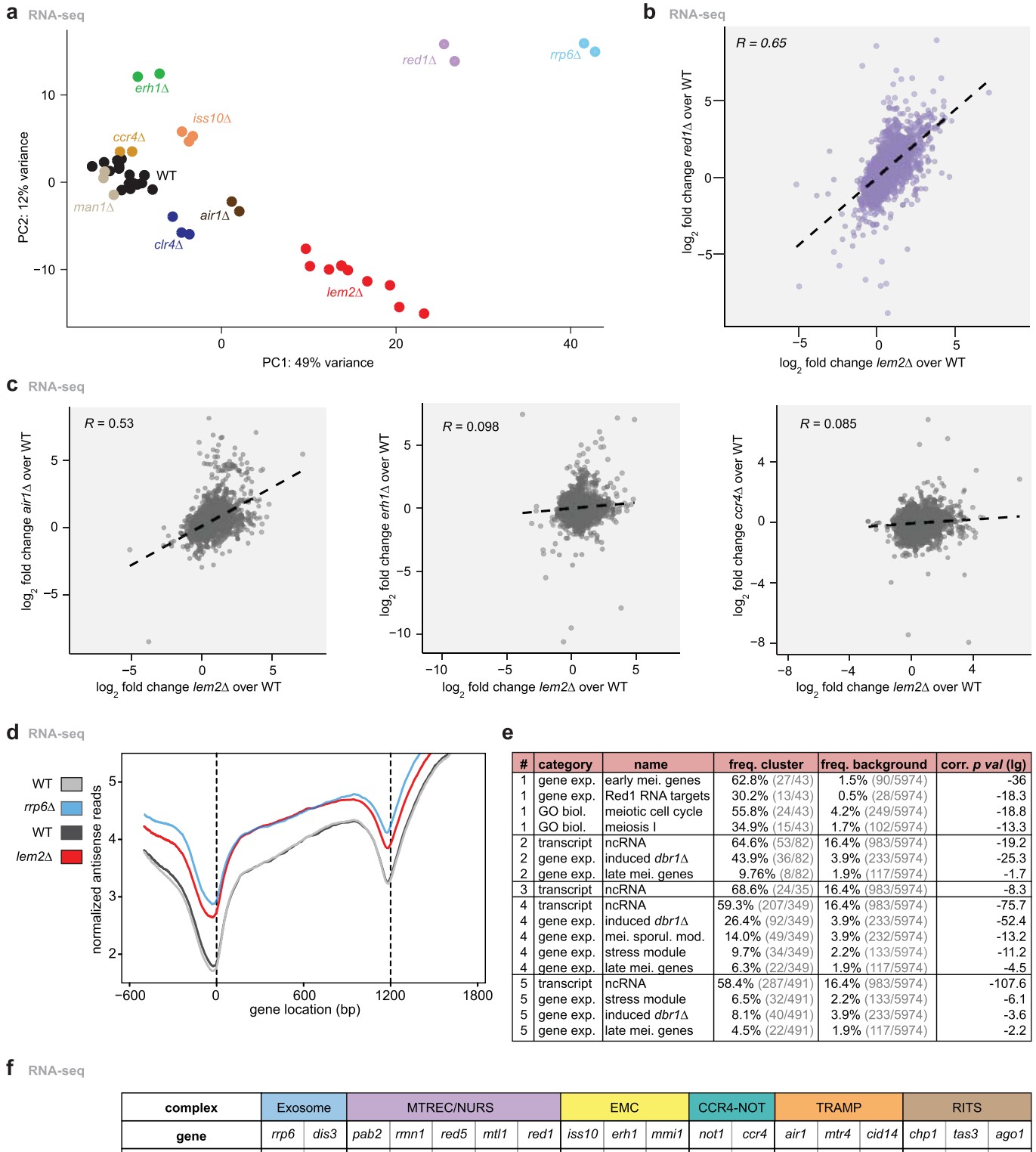

**Extended Data Fig. 2 | Lem2 cooperates with different exosome factors. a**, PCA plot from the RNA-seq libraries generated of the indicated strains. Each dot represents a biological replicate. **b**, **c**, Scatterplots of genome-wide log₂ fold change expression from transcripts in *red1Δ* vs *lem2Δ* (**b**) or *air1Δ* vs *lem2Δ*, *erh1Δ* vs *lem2Δ* and *ccr4Δ* vs *lem2Δ* (**c**), each of them relative to WT. The linear regression line is depicted together with the Pearson correlation coefficient value (*R*). **d**, Genome-wide plot showing the log₂ coverage of all annotated antisense mRNAs (Y axis, normalized antisense reads). All units are normalized to 1200 bp (X axis, gene location). The colors depict different strains (see legend on the left). **e**, Table with selected results from gene list enrichment analysis from the clusters (#) 1-5 from Fig. 2b. The Bähler Lab *AnGeLi* tool with a two-tailed Fisher's exact test and a false discovery rate of 0.05 was used for this analysis (Ref. [46]). Freq., frequency; corr., corrected; gene exp., gene expression; GO biol., Gene Ontology biological process; mei., meiotic; mid., middle, sporul. mod., sporulation module. **f**, Table showing the linear expression values of multiple exosome subunits, as shown in Fig. 2a in *lem2Δ* cells. Data were retrieved from RNA-seq analyses and are shown as log₂ fold change of *lem2Δ* over WT.

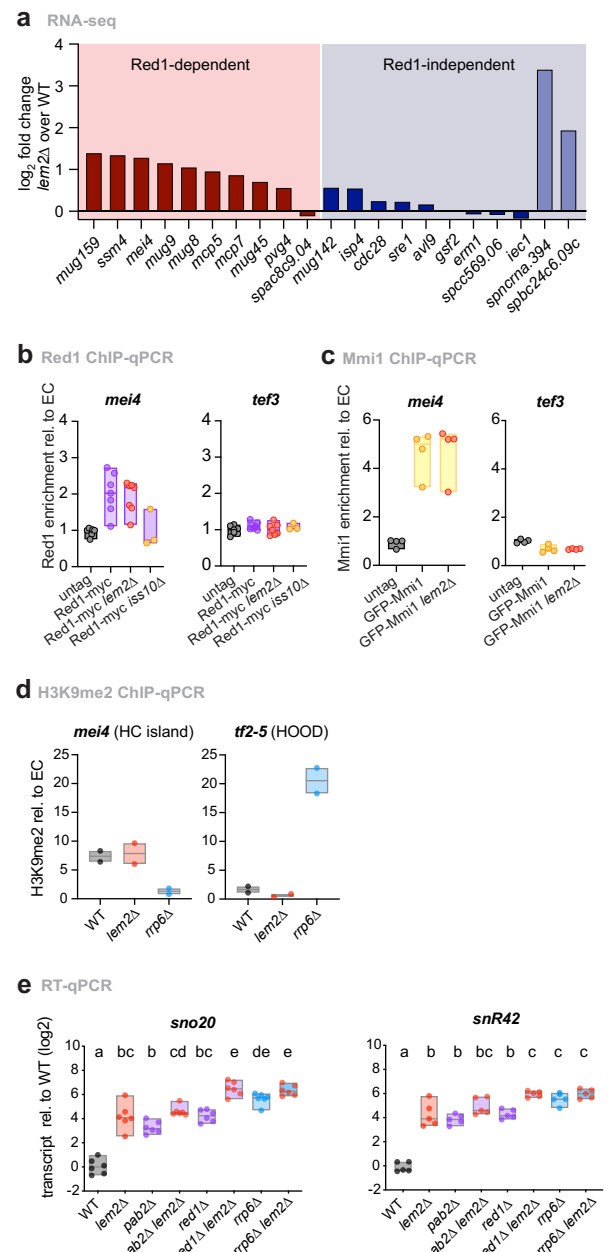

**Extended Data Fig. 3 | Lem2 collaborates with distinct RNA degradation pathways. a**, Expression levels of Red1-dependent (left, red) and Red1-independent (right, blue) islands in *lem2Δ* cells. Gene names are indicated below the graph. Data were retrieved from RNA-seq analyses and are shown as log$_2$ fold change of *lem2Δ* over WT. **b**, ChIP-qPCR analysis of Red1-Myc enrichment at *mei4+* and *tef3+* in the indicated strains (*n* = 7 independent biological replicates, except sample Red1-myc *iss10Δ*, *n* = 3). **c**, ChIP-qPCR analysis of GFP-Mmi1 enrichment at *mei4+* and *tef3+* in the indicated strains (*n* = 4 independent biological replicates). **d**, ChIP-qPCR analysis of H3K9me2 enrichment of *mei4+* (heterochromatin island), *tf2-5+* (HOOD) in the indicated strains (*n* = 2 independent biological replicates). For (**b-d**), data are divided by the input and normalized to euchromatin levels (*act1+*, *tef3+*, *ade2+*). **e**, *sno20* and *snR42* transcript levels quantified by RT-qPCR in the indicated strains. Data from independent biological replicates (*sno20*, *n* = 6; *snR42*, *n* = 5) are normalized to *act1* transcript levels and shown relative to WT on a log$_2$ scale. Statistical analysis was performed using one-way ANOVA, and letters denote significant differences between groups with Tukey's *post hoc* tests at *P* < 0.05. For (**b-e**), the individual replicates are shown in a floating bar plot and the line depicts the median.

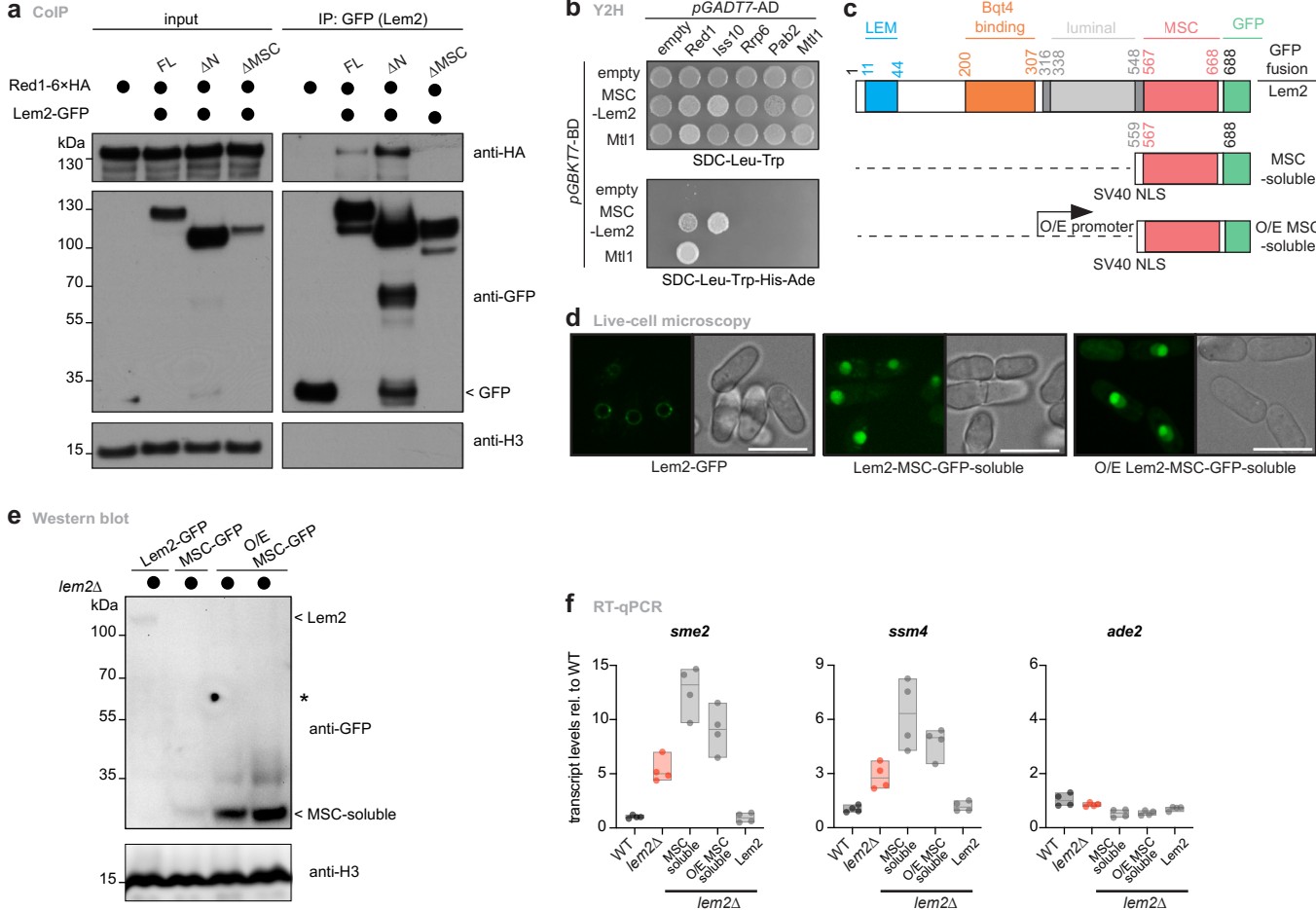

**Extended Data Fig. 4 | Lem2 interacts with the nuclear exosome through the MSC domain and functions in a location-dependent manner. a**, Co-immunoprecipitation of Red1-6xHA with Lem2-GFP WT (Lem2-GFP FL) or truncation mutants lacking the N terminal (Lem2-GFP ΔN) or MSC domain (Lem2-GFP ΔMSC). Histone H3 served as loading control. Proteins were expressed from a *pREP81* based overexpression vector transformed in *lem2*Δ strains. **b**, Y2H analysis of Red1, Iss10, Rrp6, Pab2 and Mtl1 with MSC-Lem2 or Mtl1, grown for 3 days on medium with different auxotrophies (SDC-Leu-Trp, SDC-Leu-Trp-His-Ade). Fusions with Gal4-activating domain (*pGADT7*-AD) or Gal-4-DNA-binding domain (*pGBKT7*-BD) are shown. **c**, Schematic representation of Lem2 truncation constructs. Protein domains and positions (amino acids) are indicated. All constructs were C-terminally GFP-tagged and inserted in the endogenous locus in *lem2*Δ background cells. **d**, Representative images from live-cell microscopy of the constructs in Extended Data Fig. 4c. Left panels show the GFP excited channel and right panels show the corresponding bright field images. A projection of several Z-stacks with maximum intensity is shown. For some cells, Lem2-GFP is out of plane and not visible. Scale bar = 10 μm. **e**, Immunoblot of Lem2-GFP constructs in Extended Data Fig. 4c. In here, two different promoters were used to overexpress the soluble MSC fragment (left = pGPD, right = pTEF). H3 served as loading control. Asterisk (*) indicates unspecific band. **f**, *sme2, ssm4* and *ade2* transcript levels quantified by RT-qPCR in Lem2 truncation mutants. Data from *n* = 4 independent biological replicates were normalized to *act1* transcript levels and shown relative to WT. The individual replicates are shown in a floating bar plot and the line depicts the median. For (**a**, **d**, and **e**), blots shown are representative examples of experiments reproduced at least three times.

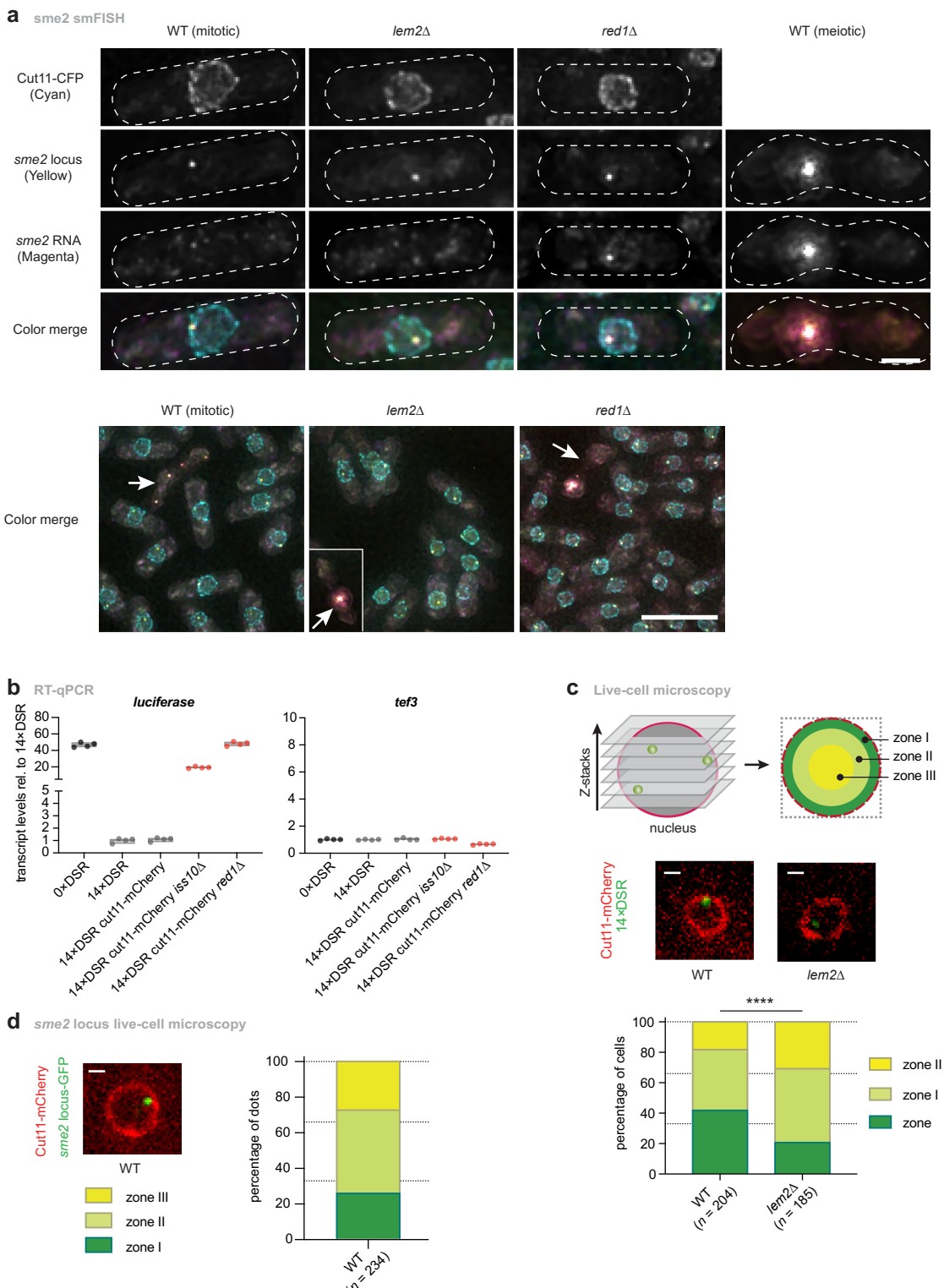

**Extended Data Fig. 5 | See next page for caption.**

**Extended Data Fig. 5 | Exosome substrates localize to the nuclear periphery. a**, Top: smFISH analysis for sme2 RNA using sequence-specific probes labeled with Quasar®570. The sme2 locus was visualized using a lacO/LacI-GFP system. Cut11-CFP was used to mark the NE. Scale bar, 2 μm. Bottom: smFISH color merge panels showing multiple cells. Arrows indicate meiotic cells, where the *sme2* RNA signal collapses to a single dot. Scale bar, 10 μm. For top and bottom: color merge images show the smFISH signal (magenta), the lacO/LacI-GFP (yellow) and Cut11-CFP (blue). **b**, Transcript levels of *luciferase* and *tef3* quantified by RT-qPCR on a strain encoding 0×DSR copies (0×DSR), 14×DSR copies (14×DSR), 14×DSR and the NE marker in a WT (14×DSR Cut11-mCherry), *iss10Δ* background (14×DSR Cut11-mCherry *iss10Δ*) or *red1Δ* background (14×DSR Cut11-mCherry *red1Δ*). Data from $n = 4$ independent biological replicates are normalized to *act1* transcript levels and shown relative to the 14×DSR strain. The individual replicates are shown in a floating bar plot and the line depicts the median. **c**, Top: Schematic representation of the live-cell imaging acquisition method. Z-stacks were acquired from cell nuclei detecting DSR dots and the NE (Cut11-mCherry). To assess localization, DSR dots were assigned to one out of three concentrical zones with equal areas within the nucleus, as measured by the distance to the NE. Bottom: Live-cell microscopy representative images of the DSR containing strain in a WT or *lem2Δ* background. Cut11-mCherry marks the NE. A single z-stack is shown. The DSR location was quantified in WT and *lem2Δ* backgrounds relative to the periphery expressed in percentage of dots. Statistical analysis was performed using $\chi^2$ test. ****, $P < 0.0001$. Scale bar = 1 μm. **d**, Left: Live-cell microscopy representative image of the *sme2+* locus (*sme2::ura4::lacOp*, *his7+::LacI-GFP*) in a WT background. Cut11-mCherry was used as a marker for the NE. A single z-stack is shown. Scale bar = 1 μm. Right: quantification of *sme2+* locus location in WT and background relative to the periphery expressed in percentage of dots. For (**c-d**), *n* denotes the number of cells counted in two independent experiments.

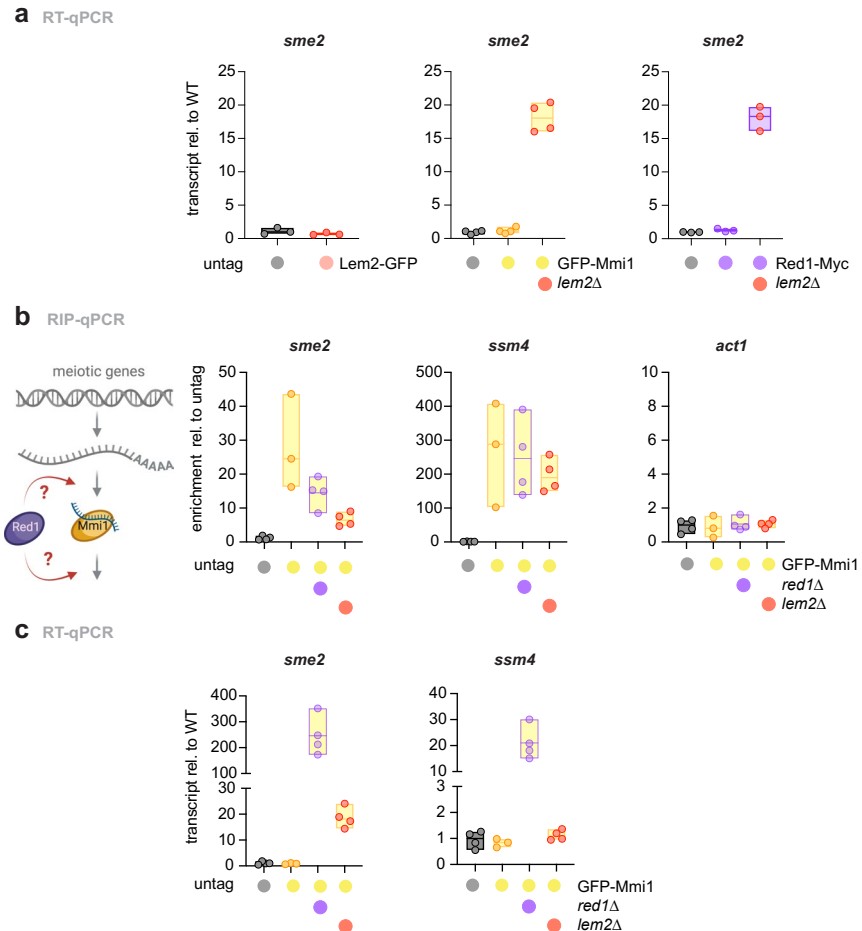

**Extended Data Fig. 6 | Lem2 regulates transcript binding by the exosome machinery. a**, Transcript levels of *sme2* quantified by RT-qPCR in the input from RIP samples in the indicated strains (Lem2-GFP, *n* = 3; GFP-Mmi1, *n* = 4, Red1-Myc, *n* = 3 independent biological replicates). Data are normalized to *act1* transcript levels and shown relative to the untagged strain. **b**, Transcript binding to GFP-Mmi1 analyzed by RIP-qPCR analysis in WT, *red1Δ* and *lem2Δ* cells. Data from *n* = 4 (except GFP-Mmi1 in WT background, *n* = 3) biological replicates are divided by the input and shown relative to the median of the untagged strain. **c**, Transcript levels of *sme2* and *ssm4* quantified by RT-qPCR in the input of the indicated RIP strains (*n* = 4; except GFP-Mmi1 in WT background, *n* = 3). Data were normalized to *act1* transcript levels and shown relative to the untagged strain. The individual replicates are shown in a floating bar plot and the line depicts the median.

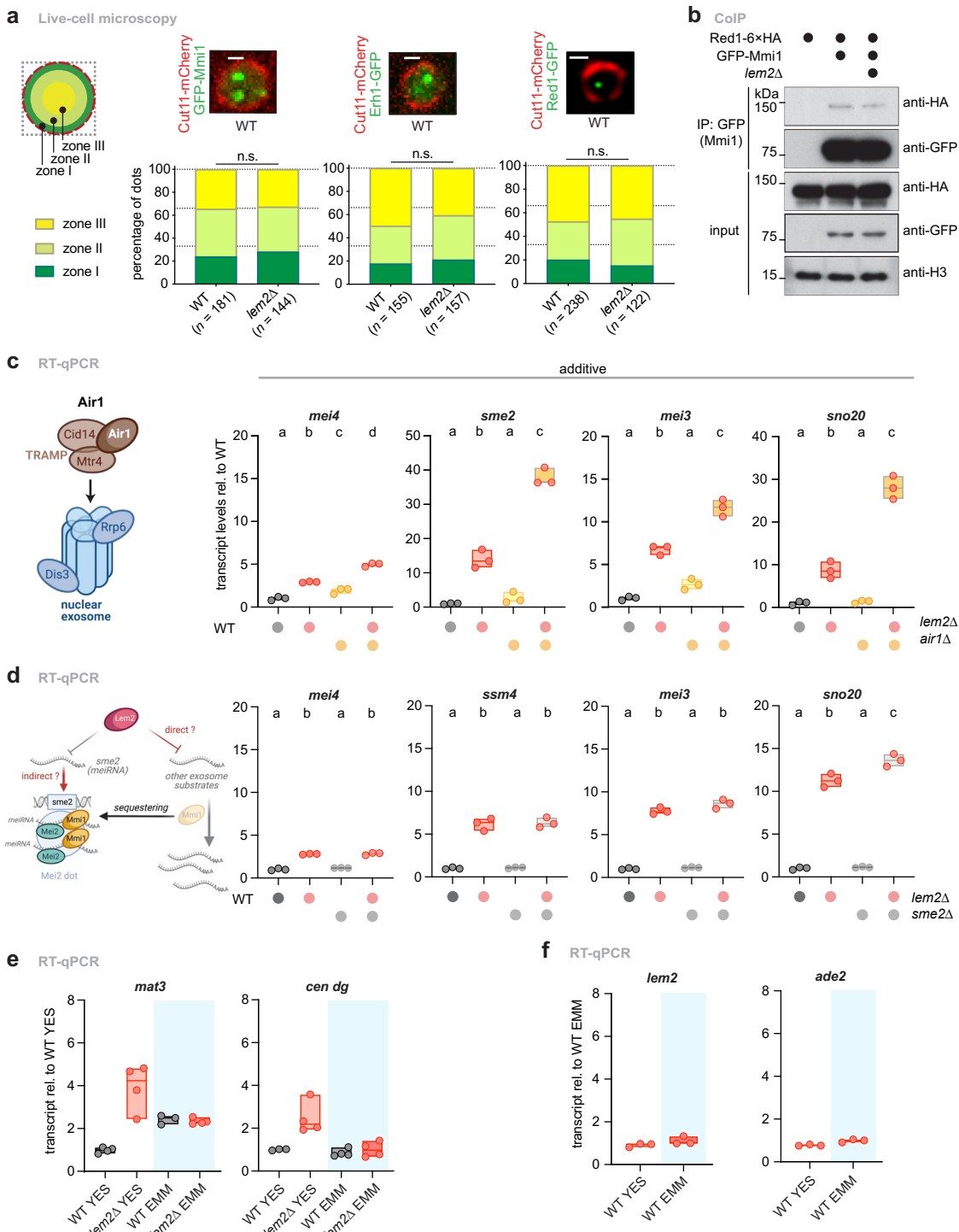

**Extended Data Fig. 7 | RNA regulation by Lem2 at the nuclear periphery occurs independently of exosome factors associated with nuclear foci. a**, Left: schematic representation of the *S. pombe* nucleus divided into three equal areas designated I-III. Right: Representative live-cell microscopy images of GFP-Mmi1, Erh1-GFP and Red1-GFP in a WT or *lem2Δ* background. Cut11-mCherry was used as a marker for the NE. A single z-stack is shown. The quantification of protein localization is shown in WT and lem2Δ backgrounds relative to the periphery expressed in percentage of dots. *n* = number of cells counted in two independent experiments. n.s. = not significant from χ2 test analysis. Scale bar = 1 μm. **b**, Co-immunoprecipitation of Red1-6×HA with GFP-Mmi1 in WT or *lem2Δ* background. H3 served as loading control. Proteins were expressed from their endogenous loci. Two independent experiments were performed with similar results. **c**, Transcript levels of *mei4*, *sme2*, *mei3* and *sno20* quantified by RT-qPCR in the indicated strains (*n* = 3). **d**, Transcript levels of *mei4*, *ssm4*, *mei3* and *sno20* quantified by RT-qPCR in the indicated strains (*n* = 3). **e**, Transcript levels of *mat3* and *cen dg* quantified by RT-qPCR in the indicated strains (*n* = 3-4). **f**, Transcript levels of *lem2* and *ade2* quantified by RT-qPCR in the indicated strains (*n* = 3). For (**c-f**), data from independent biological replicates were normalized to *act1* transcript levels and shown relative to the WT strain (**c, d**), WT in YES (**e**) or the WT in EMM (**f**). Individual replicates are shown in a floating bar plot and the line depicts the median. For (**c, d**), statistical analysis was performed using one-way ANOVA, and letters denote significant differences between groups with Tukey's *post hoc* tests at *P* < 0.05. For (**e, f**), blue shadowing indicates minimal media (EMM).

# Reporting Summary

## Statistics

For all statistical analyses, confirm that the following items are present in the figure legend, table legend, main text, or Methods section.

| n/a | Confirmed | |
|---|---|---|
| ☐ | ☒ | The exact sample size (*n*) for each experimental group/condition, given as a discrete number and unit of measurement |
| ☐ | ☒ | A statement on whether measurements were taken from distinct samples or whether the same sample was measured repeatedly |
| ☐ | ☒ | The statistical test(s) used AND whether they are one- or two-sided<br>*Only common tests should be described solely by name; describe more complex techniques in the Methods section.* |
| ☒ | ☐ | A description of all covariates tested |
| ☐ | ☒ | A description of any assumptions or corrections, such as tests of normality and adjustment for multiple comparisons |
| ☐ | ☒ | A full description of the statistical parameters including central tendency (e.g. means) or other basic estimates (e.g. regression coefficient) AND variation (e.g. standard deviation) or associated estimates of uncertainty (e.g. confidence intervals) |
| ☐ | ☒ | For null hypothesis testing, the test statistic (e.g. *F*, *t*, *r*) with confidence intervals, effect sizes, degrees of freedom and *P* value noted<br>*Give P values as exact values whenever suitable.* |
| ☒ | ☐ | For Bayesian analysis, information on the choice of priors and Markov chain Monte Carlo settings |
| ☒ | ☐ | For hierarchical and complex designs, identification of the appropriate level for tests and full reporting of outcomes |
| ☐ | ☒ | Estimates of effect sizes (e.g. Cohen's *d*, Pearson's *r*), indicating how they were calculated |

*Our web collection on statistics for biologists contains articles on many of the points above.*

## Software and code

Policy information about availability of computer code

| Data collection | Live-cell microscopy images were acquired with either a Zeiss AxioObserver Z1 confocal spinning disc microscope with an EMM-CCD camera, or an inverted Leica SP8X WLL microscope, equipped with 405 nm laser, WLL2 laser (470 - 670 nm) and acusto-optical beam splitter or a DeltaVision Elite system equipped with pco.edge 4.2 sCMOS camera.<br>For smFISH, images were acquired with a DeltaVision Elite system (GE Healthcare Inc.) equipped with pco.edge 4.2 sCMOS camera (PCO). Chromatic shifts were corrected using Chromagnon software (v0.87) using a bleed-through fluorescence image as a reference. The images were denoised by the ND-safir program 73, deconvolved using the built-in SoftWoRx software (v7.0.0), and then projected by maximum intensity projection<br>RNAseq libraries were generated with NEBNext Ultra Directional RNA Library (Illumina) and sequenced on HiSeq1500 or NextSeq 2000. Individual RT, RIP and ChIP-samples were analyzed by quantitative real-time PCR using QuantStudioTM 3 or QuantStudioTM 5 (Thermo Fisher) real-time PCR systems and processed using Thermo Fisher ConnectTM. For quantitative snoRNA analysis, a LIghtCycler LC480 (Roche) was used. |
|---|---|
| Data analysis | FiJi/ImageJ 2.1.0/1.53c (regularly updated) were used for confocal image analysis.<br>Graphpad Prism 8 was used for data plotting of qPCR data.<br>RNAseq data were analyzed using the following packages: STAR v2.7.3a; RSEM v1.3.3 ; DESeq2 v1.22.2; Tximport v1.10.1; sva v3.30.1; Bedtools v2.29.1.<br>The Bähler Lab AnGeLi tool (www.bahlerlab.info/AnGeLi) was used for the analysis of gene lists from RNAseq data. |

For manuscripts utilizing custom algorithms or software that are central to the research but not yet described in published literature, software must be made available to editors and reviewers. We strongly encourage code deposition in a community repository (e.g. GitHub). See the Nature Portfolio guidelines for submitting code & software for further information.

## Data

Policy information about availability of data

All manuscripts must include a data availability statement. This statement should provide the following information, where applicable:

- Accession codes, unique identifiers, or web links for publicly available datasets
- A description of any restrictions on data availability
- For clinical datasets or third party data, please ensure that the statement adheres to our policy

All sequencing data that support the findings of this study have been deposited in the National Center for Biotechnology Information Gene Expression Omnibus (GEO) and are accessible through the GEO Series accession number GSE174347. Full code for all NGS-related workflow is available at: https://github.com/Tsvanemden/Martin_Caballero_et_al_2021.

# Field-specific reporting

Please select the one below that is the best fit for your research. If you are not sure, read the appropriate sections before making your selection.

☒ Life sciences    ☐ Behavioural & social sciences    ☐ Ecological, evolutionary & environmental sciences

For a reference copy of the document with all sections, see nature.com/documents/nr-reporting-summary-flat.pdf

# Life sciences study design

All studies must disclose on these points even when the disclosure is negative.

| | |
|---|---|
| Sample size | No statistical methods were performed to predetermine the sample size, but the number of biological replicates for each experiment was based on similar studies (Barrales et al., 2016, PMID: 26744419; Ding et al., 2019, PMID: 31811152; Egan et al., 2014, PMID: 24713849; Shichino et al., 2018, PMID: 29424342; Sugiyama et al., 2016, PMID: 26942678; Thillainadesan et al., 2020, PMID: 32415063; Wei et al., 2021, PMID: 33574613). |
| Data exclusions | No data were excluded from the analysis unless they were classified as outliers based on the Grubbs' test using the outliers calculator provided by graphpad (https://www.graphpad.com/quickcalcs/Grubbs1.cfm) with a significant level of alpha = 0.05 |
| Replication | For all experiments, at least two independent biological replicates were performed and each replicate was reliably reproduced. |
| Randomization | For all experiments (ChIP, coIP, RIP, RNAseq, RT-qPCR, WB, Y2H), yeast cultures were grown under the same conditions and collected randomly without any bias. Microscopy data were generated from randomly selected cells (DIC channel) across several fields before analysis in the respective fluorescence channel. |
| Blinding | Blinding strategies are not relevant in the context of the experiments presented. |

# Reporting for specific materials, systems and methods

We require information from authors about some types of materials, experimental systems and methods used in many studies. Here, indicate whether each material, system or method listed is relevant to your study. If you are not sure if a list item applies to your research, read the appropriate section before selecting a response.

## Materials & experimental systems

| n/a | Involved in the study |
|---|---|
| ☐ | ☒ Antibodies |
| ☒ | ☐ Eukaryotic cell lines |
| ☒ | ☐ Palaeontology and archaeology |
| ☒ | ☐ Animals and other organisms |
| ☒ | ☐ Human research participants |
| ☒ | ☐ Clinical data |
| ☒ | ☐ Dual use research of concern |

## Methods

| n/a | Involved in the study |
|---|---|
| ☒ | ☐ ChIP-seq |
| ☒ | ☐ Flow cytometry |
| ☒ | ☐ MRI-based neuroimaging |

## Antibodies

| | |
|---|---|
| Antibodies used | - anti-HA (3F10), Rat monoclonal, Roche Cat# 11867423001, diluted 1:1,000 for western blot.<br>- anti-GFP (B-2), Mouse monoclonal, Santa Cruz Biotechnology Cat# sc-9996, diluted 1:1,000 for western blot.<br>- anti-Myc, Rabbit polyclonal, Abcam Cat# ab9106, diluted 1:2,000 for western blot.<br>- anti-H3 (1B1-B2), Mouse monoclonal, Merck Cat# MABE923, diluted 1:5,000 for western blot.<br>- anti-Rat IgG/HRP conjugate, Goat polyclonal, Merck Millipore Cat# AP136P, diluted 1:3,000 for western blot. |

- anti-Mouse IgG/HRP conjugate, Goat polyclonal, BioRad Cat# 170-6516, diluted 1:3,000 for western blot.
- anti-Rabbit IgG/HRP conjugate, Goat polyclonal, BioRad Cat# 170-6515, diluted 1:3,000 for western blot.
- GFP-Trap Agarose, Chromotek Cat# gta-100, for immunoprecipitation.
- Myc-Trap Agarose, Chromotek Cat# yta-100, for immunoprecipitation.
- anti-H3K9me2, Mouse monoclonal, Abcam Cat# ab1220, 2 µg for immunoprecipitation.
- anti-Pol II-S5P CDT 3E8,  Mouse monoclonal (generated by the lab of Dirk Eick), for immunoprecipitation.

Validation

All antibodies are commercially available and were validated by the manufactures, except for the anti-Pol II-S5P antibody, which was validated by Dirk Eick Lab.

