## [Peer Review File · Nature Structural & Molecular Biology]

Peer Review Information

Journal: Nature Structural and Molecular Biology

Manuscript Title: The inner nuclear membrane protein Lem2 coordinates RNA degradation at the nuclear periphery

Corresponding author name(s): Dr Sigurd Braun

Editorial Notes:

Reviewer Comments & Decisions:

Decision Letter, initial version:

28th Jun 2021

Dear Sigurd,

Thank you again for submitting your manuscript "The inner nuclear membrane protein Lem2 coordinates RNA degradation at the nuclear periphery". We now have received the comments from the three reviewers who evaluated your paper (appended below). In light of those reports, we remain interested in your study and would like to invite you to respond to the comments of the referees, in the form of a revised manuscript.

You will see that the reviewers appreciate the quality of the study and find the concept of a nuclear-envelope-associated RNA degradation pathway intriguing. Nevertheless, they would like to see more mechanistic insights into this pathway and its cellular role, and find that certain aspects of the study are currently preliminary. They do provide constructive suggestions for further development, which we think should be considered and included, if technically feasible.

We appreciate that the requested revisions are extensive. Given the time and effort such a revision would entail, we would of course understand if you prefer to seek more rapid publication elsewhere (especially if there is time pressure for swift publication on your side). If you do decide to submit the work elsewhere, please let us know, so that our process can be closed (or else, it would be considered dual submission). Otherwise, we expect to see your revised manuscript within 6 months. If you cannot send it within this time, please let us know. We will be happy to consider your revision so long as nothing similar has been accepted for publication at Nature Structural & Molecular Biology or published elsewhere. Should your manuscript be delayed and your article is eventually published, the received date would be that of the revised, not the original, version.

If you choose to revise your manuscript for potential publication at NSMB, please be sure to address/respond to all concerns of the referees in full in a point-by-point response and highlight all changes in the revised manuscript text file.

We are committed to providing a fair and constructive peer-review process. Do not hesitate to contact us if there are specific requests from the reviewers that you believe are technically impossible or unlikely to yield a meaningful outcome. We would also be happy to discuss a revision plan in more detail. If you are interested in this option, it would be helpful if you could send us a preliminary point-by-point response to the comments and suggestions of the reviewers.

Reporting Summary:

Please note that all key data shown in the main figures as cropped gels or blots should be presented in uncropped form, with molecular weight markers. These data can be aggregated into a single supplementary figure. While these data can be displayed in a relatively informal style, they must refer back to the relevant figures. These data should be submitted with the last revision, prior to acceptance, but you may want to start putting it together at this point.

SOURCE DATA: We also urge authors to provide, in tabular form, the data underlying the graphical representations used in figures. This is to further increase transparency in data reporting, as detailed in this editorial (<http://www.nature.com/nsmb/journal/v22/n10/full/nsmb.3110.html>). Spreadsheets can be submitted in excel format. Only one (1) file per figure is permitted; thus, for multi-paneled figures, the source data for each panel should be clearly labeled in the Excel file; alternately the data can be provided as multiple, clearly labeled sheets in an Excel file. When submitting files, the title field should indicate which figure the source data pertains to. We encourage our authors to provide source data at the revision stage, so that they are part of the peer-review process.

Deposition of deep sequencing data is mandatory, and the datasets must be released prior to or upon publication. To avoid delays in publication, dataset accession numbers must be supplied with the final accepted manuscript and appropriate release dates must be indicated at the galley proof stage. Please find the complete NRG policies on data availability at <http://www.nature.com/authors/policies/availability.html>.

While we encourage the use of color in preparing figures, please note that this will incur a charge to

partially defray the cost of printing. Information about color charges can be found at <http://www.nature.com/nsmb/authors/submit/index.html#costs>

[REDACTED]

With kind regards,
Anke

Anke Sparmann, PhD
Senior Editor
Nature Structural and Molecular Biology
ORCID 0000-0001-7695-2049

Referee expertise:

Referee #1: RNAi, heterochromatin assembly, *S. pombe*

Referee #2: RBPs, RNA degradation

Referee #3: RNA degradation, *S. pombe*

Reviewers' Comments:

Reviewer #1:

Remarks to the Author:

In this manuscript, Caballero et al describe the involvement of the nuclear envelope protein Lem2 in post-transcriptional gene silencing by the nuclear exosome RNA degradation pathway. This function of Lem2 is distinct from its role in transcriptional gene repression. The authors show that Lem2 does not directly interact with RNA substrates, but instead assists in the association of target transcripts with the components of the exosome-targeting complex MTREC. Specifically, their results show that Lem2 associates with the Red1 subunit of MTREC, which is involved in nuclear RNA processing. The Lem2-assisted RNA processing pathway acts independently of previously described nuclear bodies that contain a high concentration of RNA degradation factors and mediates silencing of exosome targets at the nuclear periphery. Finally, the authors find that Lem2-mediated regulation of target meiotic transcripts is modulated by nutritional signals.

Overall, this is an interesting study that uncovers a novel function for Lem2 in post-transcriptional gene silencing by nuclear RNA processing factors such as MTREC and the exosome. The results presented support the major conclusions. The authors shall consider addressing the following comments:

Specific comments:

1. As Lem2-dependent post-transcriptional silencing is shown to function in parallel to the MTREC-Iss10 pathway, it would be interesting to check if Lem2-dependent vs -independent transcripts have distinct features or motifs that predispose the substrate for the specific degradation pathway.
2. Fig. 3A, the interaction between Lem2 and Red1 is an important result as it suggests a direct involvement of Lem2 in RNA degradation by the exosome. However, the IP-western data presented to support this conclusion is not convincing. The authors may want to further strengthen this important result.
3. The authors suggest that the DSR-containing transcript localizes to the nuclear periphery (where Lem2 is enriched) but not to the genomic loci that produce the transcript, as investigated using the *sme2* locus (Suppl. Fig. 4C). However, previously published RNA FISH results suggest that the meiotic transcripts are associated with genomic loci in WT cells. Additional experiments are required to support their conclusion (current experiments performed using different reporter transcript/loci may not be sufficient). Also, they shall consider showing *sme2* localization in WT and *lem2Δ* cells.
4. To support the conclusions made in Fig. 4, authors could localize the reporter with respect to Lem2 and show their % co-localization when the reporter dot is at zone I.
5. Based on the data showing the localization of Red1, Erh1 and Mmi1 in WT and *lem2Δ*, the authors conclude that the majority of EMC and Red1 foci do not co-localize with the NE and may represent

independent degradation pathways that differ from the pathway regulated by Lem2. This may be oversimplifying the data as authors have not determined the extent of co-localization for these molecules and the DSR reporter dot in WT and *lem2Δ* cells.

6. In Fig. 6C, authors suggest that there is no further upregulation of the meiotic transcripts in nitrogen-starved *lem2Δ* cells. They speculate that Lem2 might be subject to post-translational modification. It would be interesting to know if there is any difference in the localization of Lem2 in nitrogen starved cells compared to normal growth conditions. Also, there seems to be a significant reduction in the transcript levels in *lem2Δ* in EMM-N when compared to YES. Further clarification of this finding is required.

7. Nuclear RNA processing factors such as Mmi1 protein (which targets MTREC) and CCR4-NOT have been shown to associate with another nuclear rim protein, Amo1 (PMID: 31883795). Perhaps this should be mentioned in the discussion section.

Minor comments:

1. Figure call out for Suppl. Fig. 1D appears before Suppl. Fig. 1B.

2. Line 363, the authors make a statement that “*sme2+* and *mei3+* transcripts were exclusively controlled by Lem2”. However, Fig. 6A shows that there is a redundant role for Erh1 in the process. The authors may want to modify this statement.

3. In the abstract, the authors state that the “Lem2-assisted pathway acts independently of nuclear bodies where exosome factors assemble, revealing that multiple spatially distinct degradation pathways exist”. Additional evidence is required to fully support this conclusion, or it should be rephrased in the abstract.

Reviewer #2:

Remarks to the Author:

This work by Caballero et al report on the new role of the nuclear membrane protein Lem2 in RNA elimination via the RNA exosome complex. They show that *lem2* mutant accumulate meiotic transcripts and ncRNAs (mostly snoRNAs) and that the RNA expression profile of *lem2* mutant overlaps with RNA-seq data from *rrp6* and *red1* mutants. They also report physical interaction between the MSC domain of Lem2 and Red1 and show that the localization of a synthetic reporter RNA that contains DSR elements is reduced in *lem2*-null cells. Functionally, Lem2 appears to play a role in facilitating the association of substrate transcripts with MTREC (Mmi and Red1), yet how Lem2 promotes the association of target transcripts with Mmi1 and Red1 remains unclear.

In general, the data presented in this work is of high quality. The idea of clustering target transcripts and the RNA exosome at the NE for temporal and spatial RNA decay is very interesting and Lem2 could be part of such pathway. However, although well done, I believed that this study in its current format remains preliminary in elucidating the mechanism by which Lem2 promotes RNA degradation. If Lem2 acts as an

anchor at the NE for MTREC, then why (and how) is Mmi1- and Red1-RNA association perturbed by the absence of Lem2? Moreover, the functional role of Lem2 in snoRNA expression also remains totally unclear. The data claiming Lem2 inactivation during early meiosis is also too preliminary in my opinion.

MAJOR POINTS:

1. The authors have used RNA-seq data to examine the possibility that expression of RNA exosome subunits and/or cofactors might be altered in *lem2*-null cells (Fig. S2F). Notably, the expression of Erh1 is almost reduced by 2-fold at the mRNA level. This can be exacerbated at the protein level as Erh1 is expressed at almost 10,000 copies per cell. I would recommend the authors to analyze a C-terminal tagged version of Erh1 in WT and *lem2* mutant to exclude the possibility that part of the *lem2* phenotype is the result of Erh1 depletion.
2. Fig. 2F and S2J. The authors claim that a non-additive effect is seen for meiotic transcript for single *lem2* mutant relative to double *lem2 pab2*, *lem2 red1*, and *lem2 rrp6* mutants, whereas an additive effect is observed for snoRNA. I am not really convinced here without any statistical analysis. For my perspective, I don't see any additive effect for snoRNAs. This needs to be clarified with statistical tests.
3. In Fig. 3, the authors examined whether the post-transcriptional function of Lem2 requires anchoring of Lem2 at the nuclear periphery. For this, they expressed a soluble version of the MSC domain of Lem2, which they have shown is sufficient for Red1 binding in Y2H. Yet, this soluble version is 5-10 times less expressed than the FL Lem2 (Fig. S3E), which could be insufficient to promote MTREC recruitment in an anchoring-independent fashion. For me, a better control would be the expression of the FL Lem2 with key mutations (or deletion) in the two transmembrane domains that would prevent anchoring at the nuclear periphery. Otherwise, it would be important to show that the MSC domain alone can recruit Red1 *in vivo*; this was only shown in Y2H. A prediction of their model is that excess soluble MSC domain in WT cells should sequester a significant fraction of Red1 away from endogenous Lem2 and lead to accumulation of meiotic transcripts.
4. While I appreciate the RNA localization assay using luciferase-DSR reporter construct to show Lem2-dependent localization of a target exosome substrate at the NE, I would encourage the authors to use smFISH (or RNAscope) to examine an endogenous target such as *ssm4* or *sme2* for example. A key issue with the reporter construct used in this study is that it contains 14 tandem DSR sequences, which is more than most endogenous Mmi1 targets (*ssm4*: 2 core and 5 augmenting motifs; *mei4*: 7 core and 1 augmenting; PMID: 22645662). What happens if the number of DSR element is cut by half in the reporter construct to reflect the number of DSR in endogenous substrates? Importantly, their model predicts that the reporter mRNA should still be at 40% in Zone I in *rrp6*, as Rrp6 is not expected to recruit the target at the NE. In fact, in *rrp6* mutant, we expect a significant increase in foci detection. In contrast, we expect that the deletion of Red1, which is thought to directly interact with Lem2 should result in reduced Zone I localization, as seen for the *lem2* mutant. These key controls need to be done to support the proposed model.
5. Post-transcriptional versus co-transcriptional: one interpretation of the RIP-qPCR data shown in Fig. 5 is that Lem2 promotes binding of the exosome targeting machinery to specific substrates post-

transcriptionally. Yet, I would argue that Red1-exosome targeting to DSR substrates by Mmi1 is a co-transcriptional process. Accordingly, ChIP-qPCR of GFP-Mmi1 and Red1-Myc to *sme2* and *ssm4* loci in WT and *lem2* mutant should be done. If the mechanism is really post-transcriptional, no change in ChIP-qPCR is expected between WT and the *lem2* mutant. In contrast, if Lem2 promotes the co-transcriptional targeting of Red1-exosome to substrate loci, a similar decrease is expected in ChIP-qPCR assays.

6. Throughout the paper, the authors refer to the fact that Lem2 represses meiotic RNAs and other ncRNAs. Yet, most of these ncRNAs in the paper refer to the accumulation of snoRNA, including sno20 and snR42. Mature snoRNAs are notoriously difficult to RT due to their highly structured regions and therefore, the observed accumulation in snoRNA coverage seen in RNA-seq is most likely snoRNA precursors. Although this is not a critical point of the paper, the authors still refer to the role of Lem2 in snoRNA “degradation or repression” throughout the study. I therefore think it would be important to resolve how Lem2 affects the expression of snoRNAs: accumulation of mature form or of a precursor species. The question being is Lem2 involved in turnover of mature snoRNA or in the mature of snoRNA precursors into mature snoRNA. This can be easily examined by northern blotting using a probe specific to the mature sno20 and snR42.

7. Line 390: “the non-additive phenotype suggests that Lem2 contributes to meiotic transcript accumulation in WT cells”. I am not sure what the authors mean by non-additive effect. To me, there was a clear time-dependent increase in *sme2* and *mei3* RNA levels in both WT and *lem2* mutant cells, which is likely the consequence of the transcriptional activation of these genes during nitrogen starvation. I disagree that these data are sufficient to support a conclusion that Lem2 is inactivated in WT cells during early meiosis to allow accumulation of meiotic transcripts. Whether Lem2 acts post-transcriptionally or co-transcriptionally also remains unclear.

MINOR POINTS

1. It would be important to show the expression of the truncated versions of Lem2 used in Fig. 3C-3D.
2. Fig. 3A. Without any positive controls for an RNA-sensitive copurification, it is not clear that the RNase and Benzonase treatment were effective.
3. Fig. 1H. Most of the Ser5-phosphorylated RNAPII is found at the 5' end of genes. Accordingly, I am not sure I would refer to Ser5P RNAPII as “elongating” RNAPII. The Ser2P mark of the CTD is most appropriate for “elongating” RNAPII.
4. p. 3, line 40-41. The TRAMP complex is not really critical for 3' end processing of snoRNAs in fission yeast. The polyadenylation-dependent maturation of 3'-extended snoRNA precursors mostly depend on the canonical Pla1 and Pab2 (PMID: 20129053).

Reviewer #3:

Remarks to the Author:

In their manuscript “The inner nuclear membrane protein Lem2 coordinates RNA degradation at the nuclear periphery”, Caballero et al. report that levels of ncRNAs and mRNAs that form part of the “Mmi1 regulon” are increased upon deletion of Lem2, a protein of the nuclear envelope with known functions in chromosome tethering and transcriptional silencing, but no reported role in promoting RNA decay. RNAs of the “Mmi1 regulon” are post-transcriptionally silenced by the RNA-binding protein Mmi1, which recognizes a defined sequence motif and promotes RNA degradation by recruiting MTREC, a nuclear exosome targeting complex that resembles human PAXT. RNA turnover may occur in granular structures where most factors involved in the pathway are concentrated, but direct evidence has not been provided. Compared to proteins that are directly involved in exosome targeting of Mmi1 regulon RNAs (*rrp6D*, *red1D*), the accumulation of meiotic RNAs upon deletion of *lem2* is less pronounced (but nevertheless clearly observable), with the exception of *sme2/meiRNA* and *sno20*, which are both strongly affected. The authors demonstrate that deletion of *lem2* has no impact on target RNA transcription (PolIII-CHIP) or heterochromatinization of target gene loci (H3K9me-CHIP), suggesting that *lem2* regulates levels post-transcriptionally. Using co-IP of truncation constructs and Y2H, the authors provide evidence for a direct interaction of Lem2 and the MTREC component Red1 and map the site of interaction to the C-terminal domain of Lem2. In the following, the authors show that deletion of *lem2* reduces the relative amount of RNA substrate that can be pulled down in RIP experiments with Mmi1 and Red1 and also decreases the likelihood of an artificial Mmi1 substrate RNA to localize to the nuclear periphery where Lem2 usually resides. The authors suggest that Lem2-dependent localization to the nuclear periphery may promote degradation of certain “Mmi1 regulon” RNAs, whereas others are more dependent on *Iss10* and *Erh1*, which facilitate the interaction of Mmi1 and MTREC and are associated with nuclear bodies. This is supported by the observation that the sensitivity of exosome target RNAs to deletion of either *lem2* or *iss10* shows an inverse relationship but can be additive when both genes are deleted (Fig 2D & 6B). The manuscript is of high scientific quality and advances our understanding of nuclear RNA surveillance by identifying the nuclear envelope protein Lem2 as a novel factor that promotes turnover of post-transcriptionally regulated RNAs at the nuclear periphery. At present, the mechanism through which Lem2 anchoring facilitates RNA turnover remains unclear, but the model where preferential recruitment to certain subnuclear locations can influence RNA fate is conceptually interesting and potentially relevant to a broader audience. The data is well presented and includes all relevant experimental detail. The study is technically sound with conclusions that are generally well supported.

Comments

I find the idea intriguing that anchoring MTREC at the nuclear periphery or in *Iss10*-dependent granules could create different environments that might be conducive to the degradation of different exosome substrates. However, as Red1 appears to be the central anchor in both cases, I am a bit puzzled that the split goes through the “Mmi1 regulon” RNAs. According to various papers cited in lines 354&355, interaction between Mmi1 and Red1 is mediated by *Iss10*, and recruitment of DSR-containing RNAs to Lem2 via Red1 should not be possible in its absence. In other words, if the literature is correct, any non-redundant, Lem2/Red1-dependent transcript should be Mmi1-independent, as is true for *sno20*. In the case of *sme2*, there are actually two isoforms, the long *meiRNA-L* packed with DSRs, and the short *meiRNA-S* without DSRs, which can be differentially affected in different mutants (see Yamashita, Noncoding RNA, 2019). The RT-qPCR primers used in this study amplify both isoforms. Did the authors check the read traces in the RNA-Seq experiments to verify whether different isoforms of *meiRNA* are

affected?

Fig 5C / lines 339/340 “deletion of *lem2*⁺ markedly reduced or abolished binding of these transcripts to both Mmi1 and Red1, revealing that Lem2 plays a critical role in the early step of RNA recognition”. I find that result somewhat surprising given that Mmi1 is known to be recruited co-transcriptionally and to be necessary for recruitment of Red1 (Tashiro et al., 2013, and others), whereas deletion of *lem2* does not interfere with Red1 recruitment to the *mei4*⁺ gene locus (Fig S2H). Moreover, according to the present study, Lem2 promotes localization of an artificial substrate RNA to the nuclear periphery (potentially through direct interaction with Red1), but not of the *sme2* gene locus, suggesting that Lem2 acts further downstream. Have the authors taken into account that the input levels of *sme2* and *ssm4* are increased upon deletion of *lem2*, i.e. can they exclude that Mmi1 levels are limiting for the amount of RNA that can be pulled down? They could a) check whether the normalization to input that they do affects the outcome of the experiment and b) include a control strain in the Mmi1-GFP pulldown that has similarly increased levels of *sme2* / *ssm4*, for example *red1D*. Alternatively, and may be more straightforward, the authors could ChIP Mmi1 in wt and *lem2D* to check whether co-transcriptional recruitment of Mmi1 is affected.

Minor points:

Line 88: Reference is missing (Wei et al., 2021)

Fig 1F, S2H&J, 6A&B: Personally, I find the letters linked to the ANOVA groups distracting rather than helpful.

Author Rebuttal to Initial comments

GENERAL RESPONSE TO REVIEWER COMMENTS¶

We thank the reviewers for their valuable comments and suggestions for additional experiments and revisions. We performed several additional experiments that further support our original conclusions. We would particularly like to highlight the following new data:

- 1. We have strengthened our conclusions that Lem2 recruits the MTREC subunit Red1 and that perinuclear localization of the MSC domain is critical for mediating its function in exosome target regulation.** We have extended our Y2H analysis and demonstrate that the MSC domain of Lem2 interacts with ZFC3H1 (human homolog of Red1), indicating that the Lem2-Red1 interaction is conserved. We also show that overexpression of the soluble MSC domain of Lem2, even at levels exceeding endogenous protein, is not sufficient to compensate for the loss of full-length, NE-bound, Lem2. These results are shown in **Figure 3b** and **Supplementary Figures 3e** and **3f**.
- 2. We provide further evidence for the perinuclear localization of RNA substrates and spatially distinct pathways in RNA degradation.** We analyzed the localization of CFP-Mmi1 in cells co-expressing the engineered DSR reporter and found that a subpopulation of Mmi1 that overlaps with the DSR dot also localizes to the NE, whereas the overall nuclear distribution of Mmi1 foci is random. These new data are shown in **Figure 4c**. Furthermore, by analyzing new transcriptome data sets for single and double mutants of *!em2*⁺ and *iss10*⁺, we find that these two proteins function in largely mutually exclusive pathways with different substrate specificities. These data are shown in **Figures 2b** and **6c**.
- 3. We provide further data consistent with our conclusion that Lem2 acts at the post-transcriptional rather than co-transcriptional level in regulating exosome targets.** We repeated and/or

performed new ChIP experiments with Red1 and Mmi1, demonstrating that their chromatin association with the genomic locus of meiotic transcripts and exosome substrate is not altered in Lem2 (**Supplementary Figure 2h** and **2i**). We provide additional data demonstrating that the binding capacity of Mmi1 is not limited when substrates accumulate in the nucleus, such as the *!em2⁺* or *red1⁺* mutants (**Supplementary Figure 6b**). These data further support the hypothesis that Lem2 promotes substrate binding by Mmi1.

- 4. We show that Lem2 plays a key role in snoRNA processing.** By analyzing snoRNA transcripts with northern blot and RT-qPCR experiments, we demonstrate that Lem2 specifically affects the 3' end processing of precursors but not the steady state level of matured snoRNAs. These results are shown in **Figure 2g** and **2h**.
- 5. We added data indicating that the function of Lem2 in turnover of meiotic transcripts is under nutritional control** by demonstrating that WT cells accumulate steady-state levels of the key meiotic regulator, the ncRNA *sme2⁺* (*meiRNA*), in minimal growth media (with or without nitrogen) to similar levels as seen in the *!em2 Δ* mutant. This result is now shown in **Figure 6d** and further supports our previous findings based on shifting cells from rich media to nitrogen starvation conditions (**Figure 6e**). However, we toned down our statement that this regulation is part of the mitosis-to-meiotic switch (and removed this conclusion from the abstract).

In addition, we thoroughly revised the text by refining our statements and adding additional references according to the reviewers' suggestions. We also substantially shortened the manuscript. Changes to the manuscript can be followed in a separate version highlighting all edits by track-changes.

Together, we believe that these changes helped to improve the quality and clarity of our manuscript and thank the reviewers again for their insightful suggestions.

Reviewer #1

In this manuscript, Caballero et al describe the involvement of the nuclear envelope protein Lem2 in post-transcriptional gene silencing by the nuclear exosome RNA degradation pathway. This function of Lem2 is distinct from its role in transcriptional gene repression. The authors show that Lem2 does not directly interact with RNA substrates, but instead assists in the association of target transcripts with the components of the exosome-targeting complex MTREC. Specifically, their results show that Lem2 associates with the Red1 subunit of MTREC, which is involved in nuclear RNA processing. The Lem2-assisted RNA processing pathway acts independently of previously described nuclear bodies that contain a high concentration of RNA degradation factors and mediates silencing of exosome targets at the nuclear periphery. Finally, the authors find that Lem2-mediated regulation of target meiotic transcripts is modulated by nutritional signals.

Overall, this is an interesting study that uncovers a novel function for Lem2 in post-transcriptional gene silencing by nuclear RNA processing factors such as MTREC and the exosome. The results presented support the major conclusions. The authors shall consider addressing the following comments:

Authors: We thank the reviewer for the assessment of our work and his/her many helpful suggestions.

Specific comments:

1. As Lem2-dependent post-transcriptional silencing is shown to function in parallel to the MTREC-Iss10 pathway, it would be interesting to *check if Lem2-dependent vs -independent transcripts have distinct features or motifs* that predispose the substrate for the specific degradation pathway.

Authors: We expanded our previous analysis of selected targets and performed a new set of RNA-seq experiments comparing the transcriptomes of *!em2LI* and *iss10LI* cells as well of the double mutant (*!em2LI iss10LI*). The results are included in the revised **Figure 2b** and the new **Figure 6c**. Interestingly, we found that genes controlled by either Lem2 or Iss10 are largely non-overlapping and that upregulation in the double mutant reflects the effects seen in the single mutants without additional increases. This is somewhat different for the two meiotic genes *ssm4⁺* and *mei4⁺*, which were more strongly upregulated in the double mutant when examined by RT-qPCR (although this increase appeared to be moderate; see our previous results in **Figure 6b**). This behavior also differs from the *!em2LI erh1LI* mutant that displays a robust synthetic increase for several meiotic genes (compare with our previous results shown in **Figure 6a**). We therefore conclude that the Lem2- and Iss10-dependent pathways are mutually exclusive, whereas some overlap exists with the Erh1-dependent pathway. These data are consistent with the role of Lem2 in promoting substrate targeting to Mmi1, which forms the EMC complex with Erh1.

Following the reviewer's suggestion, we then analyzed transcripts upregulated in either *!em2LI* or *iss10LI* for the presence of enriched motifs using the MEME suite software. Transcripts were selected based on the clustering in **Figure 6c**. These transcripts were subjected to three types of motif searches:

1. DSR-specific motif searches
2. General motif searches
3. Discriminative searches

However, none of these settings revealed motif(s) that were significantly enriched among the transcripts upregulated in one mutant, but not the other. These data are not included in the manuscript, but a detailed description can be found in the additional document attached to this author response.

2. Fig. 3A, the interaction between Lem2 and Red1 is an important result as it suggests a direct involvement of Lem2 in RNA degradation by the exosome. However, the IP-western data presented to support this conclusion is not convincing. The authors may want to further strengthen this important result.

Authors: We assume the reviewer is concerned about the significance of this result given the weak interaction, as demonstrated by the faint band for Red1-HA in the co-immunoprecipitated material. We believe that only a fraction of Red1 interacts with Lem2, given that Red1 foci do not localize to the nuclear periphery and the majority of Red1 seems to be associated with Iss10 (which acts independently of Lem2). We therefore tested whether the interaction between Lem2 and Red1 is increased in the absence of Iss10. When we performed co-IP experiments with Lem2-GFP and Red1-HA in *iss10LI* cells, multiple independent experiments show increased Red1 in the co-IPed material. However, this increase was very small (**Figure R1**, see below) and we therefore decided to not include these data in the revised manuscript.

Figure R1: In vivo association of Red1-6HA with Lem2-GFP in cells lacking Iss10. Co-IP Experiments were performed as described in Figure 3 of the revised manuscript.

Nonetheless, while the interaction between Lem2 and Red1 appears to be weak *in vivo*, there are several lines of experimental evidence that support the idea that this interaction is significant: (1) Deletion of the C-terminus (which comprises the MSC domain) but not the N-terminus of Lem2 abolished this interaction (**Supplementary Figure 3a**). (2) Y2H analysis also revealed that the MSC domain of Lem2 interacts specifically with Red1 (but not Rrp6, Pab2 or Mtl1). (3) The interaction between the MSC domain of Lem2 and Red1 is conserved for the human homologs, hLem2 and ZFC3H1, which is part of the human PAXT complex.

3. The authors suggest that the DSR-containing transcript localizes to the nuclear periphery (where Lem2 is enriched) but not to the genomic loci that produce the transcript, as investigated using the *sme2* locus (Suppl. Fig. 4C). However, previously published RNA FISH results suggest that the meiotic transcripts are associated with genomic loci in WT cells. Additional experiments are required to support their conclusion (current experiments performed using different reporter transcript/loci may not be sufficient). Also, they shall consider showing *sme2* localization in WT and *lem2* Δ cells.

Authors: Indeed, during meiotic prophase, *sme2* (*mei*RNA) RNA localizes with its genomic locus when forming the Mei2 dot structure (Ding et al., Science 2012). However, the situation seems to be more complex in vegetative cells: Yamashita and co-workers reported that Mmi1 often form multiple foci and one of the foci co-localizes with the genomic *sme2*⁺ locus (Shichino et al., Open Biol. 2014). As further shown in this study, co-localization with the *sme2*⁺ locus required the presence of DSR elements within the *sme2* RNA, suggesting that the RNA is also present at the genomic locus, although endogenously expressed *sme2* RNA (visualized by MS2-bound GFP) could not be detected. Conversely, overexpression of *sme2* RNA resulted in the formation of a single dot at its genomic locus and the collapse of Mmi1 into a single dot co-localizing with overexpressed *sme2* RNA. The latter results are reminiscent of the conditions during meiotic onset when the *sme2* RNA accumulates to high levels. In addition, Red1 and Mmi1 were reported to associate with chromatin at various loci encoding meiotic genes (e.g. *mei4*⁺, *ssm4*⁺) by several research groups, suggesting that degradation of these transcripts happens co-transcriptionally. However, when examining the subnuclear localization of these loci, Red1 was not reported to localize with the *mei4*⁺ genomic locus (Egan et al., RNA 2014). In summary, it seems that exosome factors can form multiple foci with different subnuclear localizations, and the function and location of exosome degradation is not yet fully understood.

We followed the reviewer's suggestion and performed single-molecule (sm) FISH with *sme2* in WT and *!em2L1* cells. To avoid any non-physiological effects caused by *sme2* RNA overexpression in vegetative cells (i.e., induction of haploid meiosis), we performed smFISH with endogenously expressed *sme2*⁺. As positive controls for the detection of the *sme2* RNA signal, we used either mitotically growing cells lacking Red1 or WT cells undergoing meiosis (in which the Mei2 dot should be present). These results are shown in the new **Supplementary Figure 4a**. However, while *sme2* RNA was detectable in *red1L1* cells, as expected, we could not detect any specific signal above background in WT and *!em2L1* (**Supplementary Figure 4a**). Unfortunately, this precludes any further analysis regarding perinuclear localization of substrates beyond our previous results for the DSR reporter.

Nonetheless, the engineered DSR reporter used in our study has been extensively described by Yamashita and co-workers (Shichino et al., eLife 2018) and we believe that its behavior reflects many of the properties of endogenous substrates (e.g., Mmi1-dependent foci formation, co-localization with Mmi1, and the DSR-dependent degradation by the exosome). Motivated by the reviewers' comments, we decided to study its co-localization with Mmi1 in a Lem2-dependent manner. Strikingly, we found that those Mmi1 foci co-localizing with the DSR reporter also showed a preferential localization at the NE (~ 40%) in a Lem2-dependent manner, similar to our results for the reporter itself, whereas the foci altogether showed a random distribution. These

results are shown in the new **Figure 4c**. We conclude that transcripts and exosome factors can form multiple foci that localize to different structures (i.e., chromatin or NE).

0. To support the conclusions made in Fig. 4, authors could localize the reporter with respect to Lem2 and show their % co-localization when the reporter dot is at zone I.

Authors: We appreciate the reviewer's suggestion. However, by confocal microscopy, Lem2 appears to be present throughout the NE, as our previous work and studies by our collaborator, Yasushi Hiraoka, have shown. Hence, localization of the DSR reporter in zone I (i.e., overlapping with the NE) would also indicate 'co-localization' with Lem2 and not provide any additional information.

1. Based on the data showing the localization of Red1, Erh1 and Mmi1 in WT and *lem2Δ*, the authors conclude that the majority of EMC and Red1 foci do not co-localize with the NE and may represent independent degradation pathways that differ from the pathway regulated by Lem2. This may be oversimplifying the data as authors have not determined the extent of co-localization for these molecules and the DSR reporter dot in WT and *lem2Δ* cells.

Authors: We thank the reviewer for pointing this out. We agree with the reviewer that this is a feasible experiment, which we have now performed. Please, see our response to the main point 3.

2. In Fig. 6C, authors suggest that there is no further upregulation of the meiotic transcripts in nitrogen-starved *lem2Δ* cells. They speculate that Lem2 might be subject to post-translational modification. It would be interesting to know if there is any difference in the localization of Lem2 in nitrogen starved cells compared to normal growth conditions. Also, there seems to be a significant reduction in the transcript levels in *lem2Δ* in EMM-N when compared to YES. Further clarification of this finding is required.

Authors: Previous studies by our co-authors (Yasushi Hiraoka, Yasuhiro Hirano, Yasuha Kinugasa) did not reveal any changes under different growth conditions. Nonetheless, we repeated these experiments and analyzed cells grown in YES, EMM or EMM-N but also found no differences in the localization of Lem2 (see **Figure R2**). This data is consistent with the stable protein levels observed (Tange et al., Genes Cells 2016, fig. 4c). Since these data confirmed published data, we did not include them in the revised manuscript.

Regarding changes in transcript levels using *lem2Δ* cells grown in different media, we noticed that a similar phenomenon has been observed for *red1Δ* cells (Shichino et al., PLOS Genetics 2020, compare Figure 2E with 4A, relative decrease from ~20-fold relative to WT in rich media to 3-fold in minimal media). The reason for this is not quite clear but this does not seem to be mutant-specific.

Figure R2: Localization of Lem2-GFP under different growth conditions.

3. Nuclear RNA processing factors such as Mmi1 protein (which targets MTREC) and CCR4-NOT have been shown to associate with another nuclear rim protein, Amo1 (PMID: 31883795). Perhaps this should be mentioned in the discussion section.

Authors: We thank the reviewer for this suggestion. We have added the reference and this statement to the discussion. [line 390, Ref #39]

Minor comments:

1. Figure call out for Suppl. Fig. 1D appears before Suppl. Fig. 1B.

Authors: We thank the reviewer for pointing this out. Figure call outs are now all in the correct order

2. Line 363, the authors make a statement that “*sme2+* and *mei3+* transcripts were exclusively controlled by Lem2”. However, Fig. 6A shows that there is a redundant role for Erh1 in the process. The authors may want to modify this statement.

Authors: Thanks for pointing out this discrepancy. While we sought to emphasize the distinct role of Lem2 with regard to *Iss10*, we realized that the expression “exclusively controlled by Lem2” is indeed misleading. We changed this statement into “...*sme2+* and *mei3+*, whose expression levels were unaltered in *iss10Δ* alone or in combination”.

3. In the abstract, the authors state that the “Lem2-assisted pathway acts independently of nuclear bodies where exosome factors assemble, revealing that multiple spatially distinct degradation pathways exist”. Additional evidence is required to fully support this conclusion, or it should be rephrased in the abstract.

Authors: We have now included experimental evidence revealing largely mutually exclusive genome-wide transcription patterns for *lem2Δ* and *iss10Δ* (new **Figure 6c**) and that a subpopulation of Mmi1 foci overlapping with the DSR reporter localize to the nuclear periphery (new **Figure 4c**). Taking these new data into consideration, we have refined the statement in the abstract “This pathway acts largely independently of nuclear bodies...”.

Reviewer #2:

This work by Caballero et al report on the new role of the nuclear membrane protein Lem2 in RNA elimination via the RNA exosome complex. They show that *lem2* mutant accumulate meiotic transcripts and ncRNAs (mostly snoRNAs) and that the RNA expression profile of *lem2* mutant overlaps with RNA-seq data from *rrp6* and *red1* mutants. They also report physical interaction between the MSC domain of Lem2 and Red1 and show that the localization of a synthetic reporter RNA that contains DSR elements is reduced in *lem2*-null cells. Functionally, Lem2 appears to play a role in facilitating the association of substrate transcripts with MTREC (Mmi and Red1), yet how Lem2 promotes the association of target transcripts with Mmi1 and Red1 remains unclear.

In general, the data presented in this work is of high quality. The idea of clustering target transcripts and the RNA exosome at the NE for temporal and spatial RNA decay is very interesting and Lem2 could be part of such pathway. However, although well done, I believed that this study in its current format remains preliminary in elucidating the mechanism by which Lem2 promotes RNA degradation. If Lem2 acts as an anchor at the NE for MTREC, then why (and how) is Mmi1- and Red1-RNA association perturbed by the absence of Lem2? Moreover, the functional role of Lem2 in snoRNA expression also remains totally unclear. The data claiming Lem2 inactivation during early meiosis is also too preliminary in my opinion.

We thank the reviewer for the positive comments about the quality of our work and the constructive critique.

MAJOR POINTS:

1. The authors have used RNA-seq data to examine the possibility that expression of RNA exosome subunits and/or cofactors might be altered in *lem2*-null cells (Fig. S2F). Notably, the expression of Erh1 is almost reduced by 2-fold at the mRNA level. This can be exacerbated at the protein level as Erh1 is expressed at almost 10,000 copies per cell. I would recommend the authors to analyze a C-terminal tagged version of Erh1 in WT and *lem2* mutant to exclude the possibility that part of the *lem2* phenotype is the result of Erh1 depletion.

Authors: We re-analyzed the RNAseq data by RT-qPCR, which revealed no difference for *erh1*⁺ transcript levels in *!em2L1* compared to WT cells (see below **Figure R3**). We also did not observe any changes in expression levels by live-cell imaging using Erh1-GFP. Hence, we believe that Lem2 does not affect protein expression of Erh1. Please, also note that the transcription profiles of *erh1*Δ and *!em2*Δ (**Figure 2b**) are very different, which we think makes it rather unlikely that the transcriptional upregulation seen at the genome-wide level in *!em2L1* is caused by reduced Erh1 expression.

[UNABLE TO TRANSFER FIGURE]

2. Fig. 2F and S2J. The authors claim that a non-additive effect is seen for meiotic transcript for single *lem2* mutant relative to double *lem2 pab2*, *lem2 red1*, and *lem2 rrp6* mutants, whereas an additive effect is observed for snoRNA. I am not really convinced here without any statistical analysis. For my perspective, I don't see any additive effect for snoRNAs. This needs to be clarified with statistical tests.

Authors: In order to analyze whether single and double mutants differ with respect to transcriptional upregulation, we used ANOVA and Tukey's *post-hoc* test, as indicated in the figure legend. The letters indicate whether there is, or not, a statistical difference (two letters indicate that there is a trend but no statistically significant difference). For *sme2*⁺ and *ssm4*⁺, we don't find a significant difference between *pab2L1*, *red1L1* or *rrp6L1* and the corresponding double mutant (bc vs. cd; de vs. df; f vs. ef; see **Figure 2f**). For *sno20*⁺ and *snR42*⁺, however, we find a difference between *red1L1* and *red1L1 !em2L1* (*sno20*⁺: bc vs e; *snR42*⁺: b vs. c) but not for the combinations with *pab2L1* and *rrp6L1*. We apologize for not stating this more precisely. We have refined our statement to make the distinct regulation of snoRNAs by Red1 and Lem2 clear.

3. In Fig. 3, the authors examined whether the post-transcriptional function of Lem2 requires anchoring of Lem2 at the nuclear periphery. For this, they expressed a soluble version of the MSC domain of Lem2, which they have shown is sufficient for Red1 binding in Y2H. Yet, this soluble version is 5-10 times less expressed than the FL Lem2 (Fig. S3E), which could be insufficient to promote MTREC recruitment in an anchoring-independent fashion. For me, a better control would be the expression of the FL Lem2 with key mutations (or deletion) in the two transmembrane domains that would prevent anchoring at the nuclear periphery. Otherwise, it would be important to show that the MSC domain alone can recruit Red1 in vivo; this was only shown in Y2H. A prediction of their model is that excess soluble MSC domain in WT cells should sequester a significant fraction of Red1 away from endogenous Lem2 and lead to accumulation of meiotic transcripts.

Authors: We followed the reviewer's advice and generated a construct lacking both TMDs as well as the luminal part. Somewhat unexpectedly, this construct still localized to the NE and complemented the phenotype of *!em2L1* (data not shown). We assume that redundant NE anchoring exists, which might be mediated by the interaction of Lem2 with Bqt4 (Hirano et al., Genes Cells 2018).

We also tested the second strategy suggested by the reviewer, and overexpressed the soluble MSC domain using the strong constitutive TEF promoter in *lem2L1* cells. This resulted in robust expression levels (new **Supplementary Figure 3e**), clearly exceeding the levels of the endogenously expressed MSC or full-length Lem2. However, like endogenous MSC-GFP, even overexpressed MSC-GFP did not rescue the *lem2L1* (see revised **Supplementary Figure 3e**). Hence, we conclude that perinuclear localization of the MSC domain is critical for the function of Lem2 in controlling meiotic transcripts, as we previously reported for heterochromatic transcripts (Barrales et al., Genes Dev. 2016).

4. While I appreciate the RNA localization assay using luciferase-DSR reporter construct to show Lem2-dependent localization of a target exosome substrate at the NE, I would encourage the authors to use smFISH (or RNAscope) to examine an endogenous target such as *ssm4* or *sme2* for example. A key issue with the reporter construct used in this study is that it contains 14 tandem DSR sequences, which is more than most endogenous Mmi1 targets (*ssm4*: 2 core and 5 augmenting motifs; *mei4*: 7 core and 1 augmenting; PMID: 22645662). What happens if the number of DSR element is cut by half in the reporter construct to reflect the number of DSR in endogenous substrates? Importantly, their model predicts that the reporter mRNA should still be at 40% in Zone I in *rrp6*, as Rrp6 is not expected to recruit the target at the NE. In fact, in *rrp6* mutant, we expect a significant increase in foci detection. In contrast, we expect that the deletion of Red1, which is thought to directly interact with Lem2 should result in reduced Zone I localization, as seen for the *lem2* mutant. These key controls need to be done to support the proposed model.

Authors: We followed the reviewer's advice and performed smFISH with *sme2* in WT and *lem2L1* cells (see also response to Reviewer #1). As positive controls for the detection of the *sme2* RNA signal, we used mitotically growing cells lacking Red1 and meiotic WT cells. The results are shown in the new **Supplementary Figure 4a**. However, we could not detect any specific signals above background in WT and *lem2L1*, suggesting that *sme2* RNA steady state levels are too low in these strains to be detected by smFISH (consistent with results from a previous study using an MS2-loop/GFP strategy; Yamashita et al., Open Biol. 2014). Unfortunately, this precludes any further analysis regarding perinuclear localization of substrates beyond our previous results for the DSR reporter.

Nonetheless, the engineered DSR reporter generated by Yamashita and co-workers (Shichino et al., eLife 2018) reflects many of the properties of endogenous substrates (e.g., Mmi1-dependent foci formation, co-localization with Mmi1, and DSR-dependent degradation by the exosome pathway). While the original study by Shichino et al. showed that reporters with less than 8 DSR elements could not be detected, we note that *sme2* also contains a large number of DSR motifs (13 core and 12 augmented motifs). It may also be noteworthy that *sme2* transcripts are more sensitive to perturbations caused by loss of Lem2 than other exosome factors (like *lss10* or *Erh1*, see **Figure 6**), whereas transcripts with fewer DSR elements (e.g., *ssm4⁺*, *mei4⁺*) tend to behave oppositely. Moreover, we show in the new **Figure 4c** that Mmi1 foci co-localizing with the DSR reporter also show a preferential, Lem2-dependent enrichment at the NE. Thus, overall, this DSR reporter seems to recapitulate the behavior of meiotic transcripts that are controlled by Lem2

Encouraged by the reviewer's comment, we also tested what happened to the DSR reporter in mutants lacking Red1 or *lss10*. Consistent with previous results by Shichino et al., 2018, we found that loss of Red1 or *lss10* resulted in the formation of multiple DSR foci. Most cells showed more than 5 foci per nucleus, which made it challenging to quantify the localization of these dots with respect to NE enrichment (see **Figure R3**). We therefore did not pursue this further.

Figure R4: Formation and number of foci for the DRS reporter in different mutant strains. Live-cell imaging experiments were performed as described for Figure 4 in the revised manuscript.

5. Post-transcriptional versus co-transcriptional: one interpretation of the RIP-qPCR data shown in Fig. 5 is that Lem2 promotes binding of the exosome targeting machinery to specific substrates post-transcriptionally. Yet, I would argue that Red1-exosome targeting to DSR substrates by Mmi1 is a co-transcriptional process. Accordingly, ChIP-qPCR of GFP-Mmi1 and Red1-Myc to *sme2* and *ssm4* loci in WT and *lem2* mutant should be done. If the mechanism is really post-transcriptional, no change in ChIP-qPCR is expected between WT and the *lem2* mutant. In contrast, if Lem2 promotes the co-transcriptional targeting of Red1-exosome to substrate loci, a similar decrease is expected in ChIP-qPCR assays.

Authors: We thank the reviewer for this helpful suggestion, as this approach allows us to further separate post-transcriptional from co-transcriptional functions of Lem2. Hence, we performed ChIP in WT and *lem2L1* cells with GFP-Mmi1 at the *mei4⁺* locus, where we observe robust enrichment in WT cells. We also repeated ChIP with Red1-myc (which we had already included in the first version). Intriguingly, lack of Lem2 did not affect Mmi1 or Red1 binding, whereas Red1 association was reduced in *iss10L1* cells (which we used as a positive control). These results are shown in the **Supplementary figures 2h** and **2i**. Thus, taking into account our RIP results, we argue that Lem2 acts at the post-transcriptional (promoting RNA recruitment) but not at the co-transcriptional level (mediating chromatin binding), which is consistent with our additional results (e.g., lack of Pol II enrichment at meiotic genes).

6. Throughout the paper, the authors refer to the fact that Lem2 represses meiotic RNAs and other ncRNAs. Yet, most of these ncRNAs in the paper refer to the accumulation of snoRNA, including sno20 and snR42. Mature snoRNAs are notoriously difficult to RT due to their highly structured regions and therefore, the observed accumulation in snoRNA coverage seen in RNA-seq is most likely snoRNA precursors. Although this is not a critical point of the paper, the authors still refer to the role of Lem2 in snoRNA “degradation or repression” throughout the study. I therefore think it would be important to resolve how Lem2 affects the expression of snoRNAs: accumulation of mature form or of a precursor species. The question being is Lem2 involved in turnover of mature snoRNA or in the mature of snoRNA precursors into mature snoRNA. This can be easily examined by northern blotting using a probe specific to the mature sno20 and snR42.

Authors: We are grateful to the reviewer for suggesting analyzing this unresolved role of Lem2 in snoRNA processing. We have performed RT-qPCR using oligonucleotides that discriminate between the precursor and mature form of *sno20⁺* and *snR42⁺*. Indeed, we found that only the precursor form accumulated in the *lem2L1* mutant, like in *red1L1* and *rrp6L1* cells, whereas levels of the mature form were unaffected. These results were confirmed using northern blots, and are shown in the new **Figures 2g** and **2h**. We have also changed the text throughout the manuscript and refer to 3' processing of snoRNA precursors when describing the role of Lem2 in the context of *sno20⁺* and *snR42⁺*.

7. Line 390: “the non-additive phenotype suggests that Lem2 contributes to meiotic transcript accumulation in WT cells”. I am not sure what the authors mean by non-additive effect. To me, there was a clear time-dependent increase in *sme2* and *mei3* RNA levels in both WT and *lem2* mutant cells, which is likely the

consequence of the transcriptional activation of these genes during nitrogen starvation. I disagree that these data are sufficient to support a conclusion that Lem2 is inactivated in WT cells during early meiosis to allow accumulation of meiotic transcripts. Whether Lem2 acts post-transcriptionally or co-transcriptionally also remains unclear.

Authors: We conducted additional experiments under different growth conditions, showing that deletion of *lem2⁺* does not increase the transcript level of *sme2⁺* in WT cells under stress conditions (EMM, EMM-N), in contrast to optimal growth conditions (YES). These results are shown in the new **Figure 6d** and support our previous results from the time course experiment during meiosis induction. We would argue, despite some experimental variations, that *sme2⁺* transcript levels do not rise in *lem2Δ* during the time course or exceed the level in rich media; hence, we referred to this as ‘non-additive’ effect. However, we have toned down our conclusions with respect to Lem2 regulation playing a critical function during meiotic onset and no longer mention such a role in the revised abstract.

MINOR POINTS

1. It would be important to show the expression of the truncated versions of Lem2 used in Fig. 3C-

3D. Authors: We show these data in new **Figure 3d** and **Supplementary Figure 3e**.

2. Fig. 3A. Without any positive controls for an RNA-sensitive copurification, it is not clear that the RNase and Benzonase treatment were effective.

Authors: We refined the procedure and repeated the experiments, performing RNase and Benzonase treatment after immunoprecipitation when the material is bound to beads and non-specific material has been removed by wash steps. This procedure reduces the possibility of non-specific factors in the initial lysate influencing the results. We did not observe any changes when performing the digestion after the IP. The new blot is shown in **Figure 3a**.

3. Fig. 1H. Most of the Ser5-phosphorylated RNAPII is found at the 5' end of genes. Accordingly, I am not sure I would refer to Ser5P RNAPII as “elongating” RNAPII. The Ser2P mark of the CTD is most appropriate for “elongating” RNAPII.

Authors: The reviewer is correct, we apologize for the mistake. We have removed “elongating form” from this sentence and simply refer to as “[ChIP-qPCR] with Ser5-phosphorylated RNA polymerase II (Pol II-S5P).”

4. p. 3, line 40-41. The TRAMP complex is not really critical for 3' end processing of snoRNAs in fission yeast. The polyadenylation-dependent maturation of 3'-extended snoRNA precursors mostly depend on the canonical Pla1 and Pab2 (PMID: 20129053).

Authors: We thank the reviewer for pointing this out. We changed the text accordingly [line 34]: “Fission yeast TRAMP plays only a minor role in the elimination of CUTs and small nucleolar RNA (snoRNA) processing and rather contributes to the degradation of antisense transcripts...”. We refer to the prominent roles of Pla1 and Pab2 in other parts of the manuscript [line 196] citing this reference (Ref #15).

Reviewer #3:

In their manuscript “The inner nuclear membrane protein Lem2 coordinates RNA degradation at the nuclear periphery”, Caballero et al. report that levels of ncRNAs and mRNAs that form part of the “Mmi1 regulon” are increased upon deletion of Lem2, a protein of the nuclear envelope with known functions in chromosome tethering and transcriptional silencing, but no reported role in promoting RNA decay. RNAs of the “Mmi1 regulon” are post-transcriptionally silenced by the RNA-binding protein Mmi1, which recognizes a defined sequence motif and promotes RNA degradation by recruiting MTREC, a nuclear exosome targeting complex that resembles human PAXT. RNA turnover may occur in granular structures where most factors involved in the pathway are concentrated, but direct evidence has not been provided. Compared to proteins that are directly involved in exosome targeting of Mmi1 regulon RNAs (*rrp6D*, *red1D*), the accumulation of meiotic RNAs upon deletion of *lem2* is less pronounced (but nevertheless clearly observable), with the exception of *sme2/meiRNA* and *sno20*, which are both strongly affected. The authors demonstrate that deletion of *lem2* has no impact on target RNA transcription (PolII-CHIP) or heterochromatinization of target gene loci (H3K9me-CHIP), suggesting that *lem2* regulates levels post-transcriptionally. Using co-IP of truncation constructs and Y2H, the authors provide evidence for a direct interaction of Lem2 and the MTREC component Red1 and map the site of interaction to the C-terminal domain of Lem2. In the following, the authors show that deletion of *lem2* reduces the relative amount of RNA substrate that can be pulled down in RIP experiments with Mmi1 and Red1 and also decreases the likelihood of an artificial Mmi1 substrate RNA to localize to the nuclear periphery where Lem2 usually resides. The authors suggest that Lem2-dependent localization to the nuclear periphery may promote degradation of certain “Mmi1 regulon” RNAs, whereas others are more dependent on *Iss10* and *Erh1*, which facilitate the interaction of Mmi1 and MTREC and are associated with nuclear bodies. This is supported by the observation that the sensitivity of exosome target RNAs to deletion of either *lem2* or *iss10* shows an inverse relationship but can be additive when both genes are deleted (Fig 2D & 6B).

The manuscript is of high scientific quality and advances our understanding of nuclear RNA surveillance by identifying the nuclear envelope protein Lem2 as a novel factor that promotes turnover of post-transcriptionally regulated RNAs at the nuclear periphery. At present, the mechanism through which Lem2 anchoring facilitates RNA turnover remains unclear, but the model where preferential recruitment to certain subnuclear locations can influence RNA fate is conceptually interesting and potentially relevant to a broader audience. The data is well presented and includes all relevant experimental detail. The study is technically sound with conclusions that are generally well supported.

Authors: We thank the reviewer for the positive comments about the quality of our work and relevance for a broader audience.

Comments

1. I find the idea intriguing that anchoring MTREC at the nuclear periphery or in *Iss10*-dependent granules could create different environments that might be conducive to the degradation of different exosome substrates. However, as Red1 appears to be the central anchor in both cases, I am a bit puzzled that the split goes through the “Mmi1 regulon” RNAs. According to various papers cited in lines 354&355, interaction between Mmi1 and Red1 is mediated by *Iss10*, and recruitment of DSR-containing RNAs to Lem2 via Red1 should not be possible in its absence. In other words, if the literature is correct, any non-redundant, Lem2/Red1-dependent transcript should be Mmi1-independent, as is true for *sno20*. In the case of *sme2*, there are actually two isoforms, the long *meiRNA-L* packed with DSRs, and the short *meiRNA-S* without DSRs, which can be differentially affected in different mutants (see Yamashita, Noncoding RNA, 2019).

The RT-qPCR primers used in this study amplify both isoforms. Did the authors check the read traces in the RNA-Seq experiments to verify whether different isoforms of meiRNA are affected?

Authors: We looked into the tracks (IGV) and most of the reads match the “end” of the transcript, which contains most of the DSRs. The presence of DSRs and sensitivity to Lem2 perturbations seems also to correlate, since *sme2⁺* that contains 25 DSR motifs is among the most upregulated meiotic transcripts in *lem2Δ*, whereas meiotic transcript with less DSR motifs (e.g., *mei4⁺*, *ssm4⁺*) are less sensitive. Thus, we believe that the presence of DSR in target transcripts is also relevant for their regulation by Lem2. Regarding overlapping and non-overlapping functions of Lem2 and *lss10*, we would like to add that *sme2⁺* is not affected by *lss10*, while *Erh1* seems to play a minor but redundant role with Lem2. Thus, multiple pathways seem to exist, and the physiological role of exosome foci in these degradation pathways is intriguing but not yet well understood.

2. Fig 5C / lines 339/340 “deletion of *lem2⁺* markedly reduced or abolished binding of these transcripts to both Mmi1 and Red1, revealing that Lem2 plays a critical role in the early step of RNA recognition”. I find that result somewhat surprising given that Mmi1 is known to be recruited co-transcriptionally and to be necessary for recruitment of Red1 (Tashiro et al., 2013, and others), whereas deletion of *lem2* does not interfere with Red1 recruitment to the *mei4⁺* gene locus (Fig S2H). Moreover, according to the present study, Lem2 promotes localization of an artificial substrate RNA to the nuclear periphery (potentially through direct interaction with Red1), but not of the *sme2* gene locus, suggesting that Lem2 acts further downstream. Have the authors taken into account that the input levels of *sme2* and *ssm4* are increased upon deletion of *lem2*, i.e. can they exclude that Mmi1 levels are limiting for the amount of RNA that can be pulled down? They could a) check whether the normalization to input that they do affects the outcome of the experiment and b) include a control strain in the Mmi1-GFP pulldown that has similarly increased levels of *sme2* / *ssm4*, for example *red1D*. Alternatively, and may be more straightforward, the authors could ChIP Mmi1 in wt and *lem2D* to check whether co-transcriptional recruitment of Mmi1 is affected.

Authors: RNA binding levels, as examined by RIP, are indeed normalized to input levels. As these mutants affect steady state levels of these transcripts, we find it difficult to interpret data without input normalization. However, we assessed whether Mmi1 binding capacity is limited and performed RIP with Mmi1 in the *red1L1* mutant, in which transcripts levels accumulate to 20- to 200-fold over wild-type and exceed the upregulation observed in *lem2L1* cells. We found that binding to Mmi1 was not reduced in *red1L1*, including for binding of *ssm4⁺* transcripts, although transcript levels are 20-fold increased. For *sme2⁺*, we found a small decrease in binding to Mmi1-GFP in *red1L1* but not to the extent as seen for *lem2Δ*, while *sme2⁺* transcripts levels showed the inverse correlation. Thus, we conclude that binding to Mmi1 in *lem2L1* mutants is not limited. These results are shown in the new **Supplementary Figure 5b** and **5c**.

In addition, we also performed ChIP with Mmi1-GFP, similar to Red1-myc, and found no difference in their association with chromatin at the *mei4⁺* locus (see **Supplementary Figure 2h** and **2i**). Thus, while chromatin association at the genomic loci of target transcripts has been described for various exosome factors, we conclude that multiple pathways exist that act independently.

Minor

points:

1. Line 88: Reference is missing (Wei et al., 2021)

Authors: We have added this reference (Ref. #29) [line 69]

2. Fig 1F, S2H&J, 6A&B: Personally, I find the letters linked to the ANOVA groups distracting rather than helpful.

Authors: We agree that the letters linked to the ANOVA groups make the graphs busy and can be distracting. However, we think they help to interpret the results with respect to whether double mutants show a significant increase with respect to the respective single mutants or rather display a non-additive phenotype. This can be sometimes difficult to judge by only looking at the bar graphs in cases where one of the mutants (like *lem2L1*) shows a weaker phenotype than the second mutant (e.g., *red1L1* or *rrp6L1*) while the double mutant still does not show a significant increase relative to the second (i.e., stronger) mutation. We therefore decided to keep the letters in the graphs.

Decision Letter, first revision:

1st Apr 2022

Dear Dr. Braun,

Thank you for submitting your revised manuscript "The inner nuclear membrane protein Lem2 coordinates RNA degradation at the nuclear periphery" (NSMB-A44992A). It has now been seen by the original referees and their comments are below. The reviewers find that the paper has improved in revision, and therefore we'll be happy in principle to publish it in Nature Structural & Molecular Biology, pending minor revisions to satisfy the referees' final requests and to comply with our editorial and formatting guidelines.

Sincerely,

Carolina

Carolina Perdigoto, PhD
Chief Editor
Nature Structural & Molecular Biology

orcid.org/0000-0002-5783-7106

Reviewer #1 (Remarks to the Author):

The authors have addressed all my key concerns. The manuscript can be now be considered for publication in NSMB.

Reviewer #2 (Remarks to the Author):

The authors have satisfyingly addressed my many comments and requests by the addition of new data and clarifying the text in the manuscript. I therefore support publication of this study in NSMB.

Reviewer #3 (Remarks to the Author):

All points raised were satisfactorily addressed by the authors.

Final Decision Letter:

Dear Dr. Braun,

We are now happy to accept your revised paper "The inner nuclear membrane protein Lem2 coordinates RNA degradation at the nuclear periphery" for publication as a Article in Nature Structural & Molecular Biology.

As soon as your article is published, you can generate your shareable link by entering the DOI of your article here: http://authors.springernature.com/share. Corresponding authors will also receive an automated email with the shareable link

Your paper will be published online soon after we receive proof corrections and will appear in print in the next available issue. You can find out your date of online publication by contacting the production team shortly after sending your proof corrections. Content is published online weekly on Mondays and Thursdays, and the embargo is set at 16:00 London time (GMT)/11:00 am US Eastern time (EST) on the day of publication. Now is the time to inform your Public Relations or Press Office about your paper, as they might be interested in promoting its publication. This will allow them time to prepare an accurate and satisfactory press release. Include your manuscript tracking number (NSMB-A44992B) and our journal name, which they will need when they contact our press office.

About one week before your paper is published online, we shall be distributing a press release to news organizations worldwide, which may very well include details of your work. We are happy for your institution or funding agency to prepare its own press release, but it must mention the embargo date and Nature Structural & Molecular Biology. If you or your Press Office have any enquiries in the meantime, please contact press@nature.com.

Please note that *Nature Structural & Molecular Biology* is a Transformative Journal (TJ). Authors may publish their research with us through the traditional subscription access route or make their paper immediately open access through payment of an article-processing charge (APC). Authors will not be required to make a final decision about access to their article until it has been accepted. Find out more about Transformative Journals

Authors may need to take specific actions to achieve compliance with funder and institutional open access mandates. If your research is supported by a funder that requires immediate open access (e.g. according to Plan S principles) then you should select the gold OA route, and we will direct you to the compliant route where possible. For authors selecting the subscription publication route, the journal's standard licensing terms will need to be accepted, including self-archiving policies. Those licensing terms will supersede any other terms that the author or any third party may assert apply to any version of the manuscript.

Sincerely,

Carolina

Carolina Perdigoto, PhD
Chief Editor
Nature Structural & Molecular Biology
orcid.org/0000-0002-5783-7106